# LANGUAGE AGENTS WITH REINFORCEMENT LEARNING FOR STRATEGIC PLAY IN THE WEREWOLF GAME

## ABSTRACT

Agents built with large language models (LLMs) have recently achieved great advancements. However, most of the efforts focus on single-agent or cooperative settings, leaving more general multi-agent environments underexplored. We propose a new framework powered by reinforcement learning (RL) to develop strategic language agents, i.e., LLM-based agents with strategic thinking ability, for a popular language game, Werewolf. Werewolf is a social deduction game with hidden roles that involves both cooperation and competition and emphasizes deceptive communication and diverse gameplay. Our agent tackles this game by first using LLMs to reason about potential deceptions and generate a set of strategically diverse actions. Then an RL policy, which selects an action from the candidates, is learned by population-based training to enhance the agents' decision-making ability. By combining LLMs with the RL policy, our agent produces a variety of emergent strategies, achieves the highest win rate against other LLM-based agents, and stays robust against adversarial human players in the Werewolf game.

## 1 INTRODUCTION

Developing agents that are capable of logical thinking, strategic planning, and communicating with humans has been a longstanding aspiration (Wooldridge & Jennings, 1995; Goodwin, 1995) Due to the remarkable reasoning power and emergent generalization ability, large language models (LLMs) have shown great potential in constructing intelligent agents and have already led to many recent advancements (Ouyang et al., 2022; Wei et al., 2022a; OpenAI, 2023). These LLM-based agents demonstrate proficiency in solving tasks in web surfing (Nakano et al., 2021; Yao et al., 2022a), complex video games (Wang et al., 2023c; Zhu et al., 2023), and real-world applications (Ahn et al., 2022; Brohan et al., 2023). Moreover, when interacting with other players, LLM-based agents exhibit the ability to generate human-like behaviors (Park et al., 2023; Gao et al., 2023) and achieve zero-shot multi-agent cooperation (Li et al., 2023; Chen et al., 2023).

Although much progress has been made in designing LLM-based agents, most works focus on single-agent or fully cooperative tasks. Some other works (Meta et al., 2022) build language agents for more general environments but rely on predefined atomic actions. By contrast, real-world communications between humans are based on natural languages and require both cooperation and competition. Existing research efforts can be limited in the case of deploying agents in these more complex multi-agent scenarios. We consider the Werewolf game as a challenging mixed cooperative-competitive multi-agent testbed for LLM-based agents and examine their performance by playing against other LLM-based agents and human players.

Werewolf is one of the most popular social deduction games where two teams of players with hidden roles need to communicate with natural languages to discover each other's identity and eliminate their opponents. During gameplay, the Werewolves need to conceal or lie about their roles to avoid suspicions, while the Villagers aim to gather information and find hidden opponents. The game is characterized by discussions, debates, and accusations as agents try to figure out others' true identities, which requires strong communication and strategic thinking that challenge the ability of LLM-based agents. One key challenge of the Werewolf game is to identify the hidden roles from unreliable information with potential deceptions. With the existence of opponents with unknown identities, the communication between players can be uninformative or even deceptive. Agents must distinguish between truths and lies and carefully reason to deduce the true identities of other players.

Prior work on LLM-based agents for single-agent (Yao et al., 2022b) or cooperative tasks (Mandi et al., 2023) cannot handle this challenge well as their reasoning and actions are grounded in credible information and can be misled by the manipulative statements to make wrong decisions (Wang et al., 2023b). Moreover, the competitive nature of this game requires agents to employ strategically diverse actions to avoid being exploited by their opponents. If agents always adopt the same strategy, the fixed patterns in their play can be perceived and leveraged by skilled players to gain a significant advantage. For example, if the Werewolves invariably defend their accused teammates and follow their votes, the Villagers can easily reveal them through this pattern and cooperate to defeat them. This requires agents to have strong diversity in their gameplay strategies to reduce predictable behavior patterns, so that they can be less exploitable in the game. Unfortunately, existing LLM-based agents (Xu et al., 2023a) rarely consider their behavior exploitability and tend to take actions with clear strategic patterns in their play, making them vulnerable to real human players.

In this work, we propose a framework that combines LLMs and reinforcement learning (RL) to build strategic language agents, i.e., LLM-based agents with strategic thinking ability, to tackle the aforementioned challenges in the Werewolf game. Our agent uses an LLM to organize key information to reason about hidden roles and generates a diverse set of action candidates. Then we learn an RL policy by population-based training to output final actions from the candidates and achieve strong strategic play. More specifically, our agent consists of three components. The first deduction component performs reasoning over the game history. It categorizes the whole game history into a list of atomic information according to their importance and reliability and uses an LLM to deduce the hidden roles of other players. The categorized information and deduction result are used as the input for the diverse action generation component to prompt the LLM for a set of strategically diverse action candidates. These candidates provide our agent with a variety of play styles, making it possible for the agent to learn diverse and unpredictable behaviors. The last component is an RL policy to select the output actions and optimize overall decision-making performances. To make the final policy more unexploitable, we generate a pool of fixed LLM-based agents with different styles and use population-based training to further improve the RL policy by playing against itself, its past versions, and the agents in the pool.

To demonstrate the effectiveness of our framework, we perform a round-robin tournament evaluation between our agent and three other Werewolf baseline agents, where our agent consistently achieves the highest win rate. We then evaluate our agent against real humans and find it is robust to adversarial human players and achieves higher win rates than average humans in single-human settings where the rest players are all LLM-based agents. Moreover, we show that the RL policy trained with one LLM can be directly deployed to other LLMs to improve their performance in decision-making, showing zero-shot transfer capability of the RL policy. We also perform an empirical behavior analysis and find that our agent exhibits a diverse range of emergent strategies like concealment, cooperation, bluffing, and sacrificing, which are often utilized by skilled human players.

## 2 RELATED WORK

**Building agents with large language models.** There is a recent trend in developing agents with large language models (LLMs) for various domains including website environment (Nakano et al., 2021; Yao et al., 2022a; Deng et al., 2023), game and simulation (Wang et al., 2023c;a; Zhu et al., 2023; Huang et al., 2022a), real-world scenarios (Ahn et al., 2022; Brohan et al., 2023; Vemprala et al., 2023; Huang et al., 2022b), and multi-agent interaction (Park et al., 2023; Li et al., 2023; Chen et al., 2023; Mandi et al., 2023). A shared foundation of these works is to utilize LLMs for planning and decision-making. One widely used approach to improve these abilities is task decomposition. Chain-of-thought (CoT) (Wei et al., 2022b) decomposes harder tasks into simpler ones by asking the model to think step-by-step. Tree-of-thoughts (ToT) (Yao et al., 2023) extends CoT by generating multiple thoughts at each step to create a tree structure and planning by searching in the tree. Work by Gandhi et al. (2023) further combines CoT with few-shot examples to enable strategic reasoning in matrix games and negotiation games. Another line of work uses self-reflection that allows agents to reflect on previous mistakes and refine their actions (Yao et al., 2022b; Shinn et al., 2023), or uses LLMs to design reward function for training RL agents (Kwon et al., 2023; Ma et al., 2023). Our work takes a different approach by utilizing LLMs to generate candidate actions and training an RL policy to optimize decision-making.

While many LLM-based agents have been built for single-agent or cooperative scenarios, there has been limited effort in developing agents with LLMs in multi-agent mixed cooperative-competitive environments like the Werewolf game. One representative work is Cicero (Meta et al., 2022) which combines LLMs with RL to achieve human-level play in the game of Diplomacy. The main difference between Cicero and our method is that Cicero uses the RL policy to choose from a predefined action set. By contrast, the actions in our methods are natural languages generated by LLMs during the game, and the RL policy is used to choose from these actions which are not known in advance.

Another closely related work includes the concurrent study (Xu et al., 2023a) that also builds an LLM-based Werewolf agent. Their agent is purely based on LLMs and uses heuristic retrieval of key information and reflection on past experiences to enhance its ability, while our agent combines LLMs with an RL policy to further optimize performance. Some other work (Guo et al., 2023; Wang et al., 2023b) also develops pure LLM-based agents for games like Leduc Hold'em and Avalon.

**Reinforcement learning in non-cooperative games.** Applying reinforcement learning (RL) to non-cooperative games has achieved great success in the game of Go (Silver et al., 2016; 2018), poker (Moravčík et al., 2017; Brown & Sandholm, 2018; 2019), and video games (Vinyals et al., 2019; Berner et al., 2019). The most popular method that underlies these achievements is self-play and its variants (Heinrich et al., 2015; Heinrich & Silver, 2016; Hennes et al., 2020; Xu et al., 2023b), which learn a policy by training against itself and past checkpoints. Population-based training methods like policy-space response oracles (PSRO) (Lanctot et al., 2017; Muller et al., 2019) and league training (Vinyals et al., 2019) generalize self-play by maintaining a pool of different policies and training against the population. Another notable line of work is based on regret minimization techniques such as counterfactual regret minimization (CFR) (Zinkevich et al., 2007; Lanctot et al., 2009; Brown et al., 2019). DeepRole (Serrino et al., 2019) integrates deductive reasoning into CFR to solve the hidden role game named Avalon. Werewolf and Avalon are alike in that they both feature hidden roles, but Werewolf depends more heavily on natural language communication. In fact, DeepRole plays Avalon without communication and still outperforms human players. By contrast, it is almost impossible to achieve a strong play in the Werewolf game without communication.

## 3 THE WEREWOLF GAME

**Setup.** We consider a seven-player version of the Werewolf game with two Werewolves, one Seer, one Doctor, and three Villagers. An example of this game is shown in Fig. 1 and the detailed rules can be found in Appendix B. At the beginning of the game, each player is randomly assigned a hidden role, which divides them into the Werewolves and the Villagers. The two Werewolves know each other's identity and hence also know which players are the Villagers. Their goal is to kill every innocent player and avoid being discovered. The Villagers consist of three Villagers without ability and two special roles including the Seer and the Doctor. None of these players know the hidden roles of other players and their goal is to identify and eliminate the secret Werewolves.

**Gameplay.** The game alternates between night and day rounds, starting with the night. In the night round, everyone closes their eyes to let the Werewolves and special roles take secret actions. The Werewolves pick one player to kill. The Seer chooses one player to check if this player is a Werewolf. The Doctor chooses one player to save without knowing who is the target of the Werewolves. If the Doctor chooses the same player as the Werewolves, the player is successfully saved and no one is killed in the night round. Otherwise, the player is eliminated from the game.

In the day round, an announcement is first made to every player about who was killed or no one was killed last night. Then the remaining players take turns to speak in a discussion to express their opinion about who might be the Werewolves. Players can choose to claim or lie about their true identities, share or withhold information they have discovered, and accuse or defend other players to achieve their purposes. After all players have participated in a round of discussion, a vote is held to choose one suspicious player. Each player can vote for one player or do not vote, and the player with the most votes will be eliminated. The game then continues to the next night round until the Werewolves or the Villagers win the game.

**Winning.** The Villagers win the game if both Werewolves are eliminated. The Werewolves win the game if the number of Werewolves is equal to the number of Villagers left. The winning condition is checked after every night and day round.

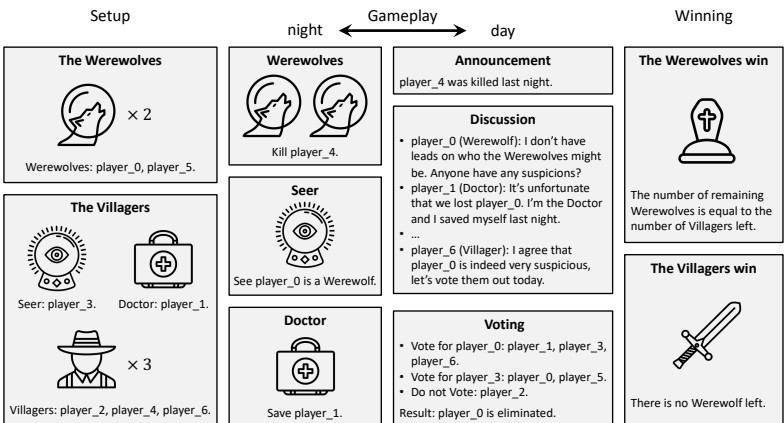

Figure 1: An example of the Werewolf game with seven players. Players are randomly assigned a hidden role and are divided into the Werewolves and the Villagers. The game alternates between night and day rounds until the Werewolves or the Villagers achieve the winning condition.

**Observations and actions.** We implement a pure text-based Werewolf game environment that does not consider external factors like the players' tone or facial expressions. The observation of each player is a text that records the current game history. This includes their ID and hidden role, their secret night actions (if any), the announcements, the discussions, and the voting results. For Werewolves, the ID and secret actions of their teammates are also in the observation. The actions of players can be divided into three types. The first is the secret actions at night including killing, seeing, and saving that choose a specific target. The second type is the statement actions during discussion which are natural languages that convey the player's opinion and information. The last type is the voting actions that can target any surviving player or choose not to vote.

## 4 STRATEGIC LANGUAGE AGENTS

The main challenges of the Werewolf game come from the adversarial opponents and ubiquitous deceptions in their claims during discussions. Players are required to deduce the hidden roles of other players from unreliable information and adopt a diverse range of actions to avoid exploitation. To achieve strong play in the game, we propose to build LLM-based agents with strategic thinking abilities with reinforcement learning (RL), which we call *strategic language agents*. Our agent uses an LLM to first distinguish credible information from potential deceptions and apply deductive reasoning to analyze the hidden roles of other players. Then the categorized information and reasoning results are used to prompt the LLM for strategically diverse action candidates. To optimize the final decision-making, an RL policy that selects the output from the candidates is learned by population-based training against itself and an opponent pool. An overview is shown in Fig. 2.

### 4.1 DEDUCTIVE REASONING

The performance of Werewolf agents is largely determined by their judgment of other players' identities. However, when faced with a large amount of mixed truth and deception, the agents can easily be misled by manipulative claims or overwhelmed by unimportant details. Consider the situation when player_3 is the Seer and they saw player_0 is a Werewolf in the first night round. This is the most important and credible information that should have a decisive impact on their reasoning and decision-making. Nevertheless, this information is surrounded by an abundance of other information like who was killed last night and the discussions of other players, making it hard for this crucial information to be discovered. This problem becomes more pronounced in the later stages of the game as agents acquire more and more information. To address this issue, we maintain an organized information record as well as a deductive reasoning result. The information record keeps key information and distinguishes truthful and deceptive statements, while the deduction result deduces the hidden role of each player and rates their reliability with an LLM.

The information record is initialized by itemizing the current observation into a list of atomic information like ["you are player_3, your role is the Seer", "you saw player_0 is a Werewolf", "player_0 says ...", ...]. These atomic pieces of information are further classified into three types including facts, potential truths, and potential deceptions. All available information except for the players' statements is included in facts, which cover established facts like the current player's role, their secret actions, the announcements, and the voting results. The statements are classified into potential

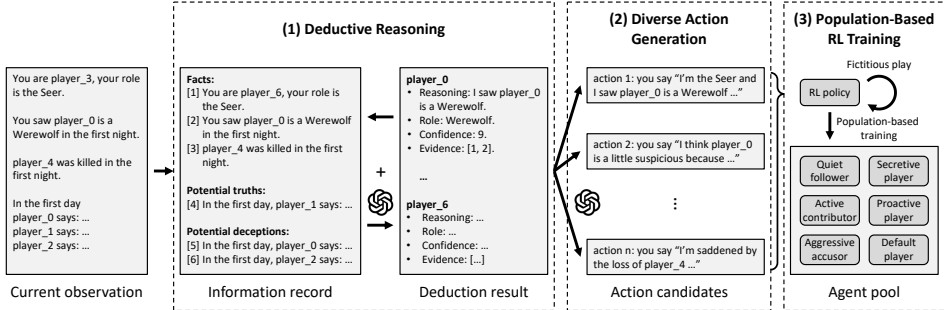

Figure 2: Overview of our agent. (1) Deductive reasoning: classify key information and apply deductive reasoning with the LLM. (2) Diverse action generation: prompt the LLM for a set of strategically diverse action candidates. (3) Population-based RL training: learn an RL policy by playing against itself, its past versions, and an agent pool.

truths or deceptions according to the players' reliability in the deduction result. In our implementation, the players' reliability is rated on a scale from 1 to 10, and the statements of players with reliability larger than 6 are regarded as potential truths, otherwise are potential deceptions.

With the organized information record, we then prompt the LLM to deduce the hidden roles of others. For each player, LLMs are asked to generate four attributes including reasoning, role, confidence, and evidence. The reasoning attribute is an auxiliary one that explicitly shows the deduction process of LLMs, which has been widely used to improve their performance in a variety of applications such as knowledge-intensive tasks (Yao et al., 2022b) and decision making tasks (Shinn et al., 2023). The role attribute corresponds to the most likely hidden role of the specific player, and the confidence attribute is an integer ranging from 5 to 10 that rates the certainty of the current deduction, where 5 means a random guess and 10 means absolutely sure. These two attributes are then used to determine the reliability of the player. If the deduced role is Werewolf, the reliability is calculated as $11-$ confidence, otherwise, the reliability is equal to the confidence. Note that the reliability of a player deduced as a Werewolf cannot be larger than $11-5=6$, which means that the statements of this player are always classified as potential deceptions. The last evidence attribute is a list of integers citing items from the information record that support the current deduction. This is used to identify the key information that contributes to the deduction of hidden roles. If a statement is never cited as evidence for deductions, it is regarded as an uninformative item and removed from the information record.

During the gameplay, the information record and deduction result are updated alternatively using each other. When new information arrives and the agent needs to make a decision, the information record is first updated by itemizing and categorizing the new information according to the previous deduction result. Then the updated information record is used as the input to prompt the LLM for a new deduction result. This result loops back to revise the record by removing uncited items and reclassifying potential truths and deceptions. The combination of information record and deduction result transforms the raw observation into structured and informative data, which serves as the foundation for subsequent decision-making.

## 4.2 DIVERSE ACTION GENERATION

Utilizing a range of strategically diverse actions is a crucial ability in the Werewolf game due to its zero-sum property. Players usually have many possible actions to take in the same situation, but no single action leads to the optimal outcome because a fixed action can be easily discerned and exploited by adversarial players. Suppose the agent is a Werewolf and the remaining players are the agent, the Seer, and the Doctor. All players know each other's identity and the agent's only chance to win is to successfully kill a player in the coming night. If the agent always takes the same action to kill the Seer or the Doctor, an observant Doctor can remember the deterministic patterns and save the target. To achieve optimal results, the agent's best strategy is to randomly choose a player to kill. This makes the agent less exploitable and no Doctor can achieve a higher win rate than 50%.

However, given the same situation, directly using LLMs often leads to a clear preference for a specific action. In the aforementioned Seer or Doctor example, we independently prompt gpt-3.5-turbo to choose a target 10 times and find that it chooses the Doctor 9 times and the Seer only once. This shows a clear bias toward the Doctor which is inherited inevitably from the model's training data. This lack of strategic diversity is also observed in other scenarios of the game where LLMs

tend to produce conservative actions like Werewolf trying to stay unnoticed and Seer hesitating to share information, which can be leveraged by adversarial opponents to gain a significant advantage.

To enhance diversity and reduce exploitation, our agent prompts LLMs to produce a set of action candidates instead of a single action. We use the concatenation of the information record and deduction result as input and consider two ways to generate $N$ action candidates with strategic diversity. The first method is to produce all candidates in a single round by prompting LLMs to "propose $N$ diverse actions that correspond to different strategies". This takes just one inference and works well for simpler actions like the secret actions and the voting actions. For more complex actions like the statement actions in the discussion, we consider a second way that iteratively generates one action for $N$ rounds by prompting LLMs to "consider a new action that is strategically different from existing ones". By having more interactions with LLMs, the second method is empirically found to produce more diverse actions with higher quality for the statement actions. In our implementation, we use the second way to prompt statement actions for quality and use the first way to prompt the secret actions and vote actions for efficiency. We also ask LLMs to output reasonings and actions for better performance (Yao et al., 2022b). The detailed prompts can be found in Appendix C.

### 4.3 POPULATION-BASED RL TRAINING

With the diverse set of candidates at hand, the agents can choose from a variety of different actions to take. Although random sampling already leads to unpredictable play, the optimal policy in most cases is a non-uniform distribution over the candidates and depends on the game state. To optimize decision-making and achieve strategic play, we use reinforcement learning to train a policy that selects the final action from the candidates.

The main difference between our setting and classic RL environments is that our action space is a discrete set of natural languages generated by LLMs. Because our action space is not predefined, we cannot use typical policy networks that only take the state as input and produce a distribution over the fixed action set. Instead, we first convert the game state and all candidate actions from natural language to vector embeddings using LLMs. Then we adopt a self-attention (Vaswani et al., 2017) network that takes all embeddings as input to produce a distribution over the action candidates. More specifically, the game state is the concatenation of the information record and deduction result described in Section 4.1, and an action candidate is the concatenation of the reasoning and action generated by LLMs as described in Section 4.2. These natural languages are converted into vector embeddings by the LLM. We also use a vector that contains player information like ID, role, etc. and pass the vector through an MLP encoder to produce a player embedding. The player embedding and language embeddings are passed through a residual self-attention block without position embeddings, and the probability to sample an action candidate is calculated as the normalized dot-product attention between the output state embedding and the output action embedding.

To learn in this mixed cooperative-competitive game, we draw inspiration from fictitious play and following work on MARL (Heinrich et al., 2015; Hennes et al., 2020) to train the policy by playing against itself and its past checkpoints. Moreover, real-world games are usually non-transitive (Czarnecki et al., 2020) and the learned policies may cycle like Rock-Paper-Scissor. The agents can benefit from playing with a variety of teammates and opponents with different styles to achieve a higher level of play. To this end, we generate a pool of fixed LLM-based agents with diverse styles to serve as teammates and opponents. These manually-designed agents also apply the information organizing and deductive reasoning step in Section 4.1 but only generate one final action according to their predefined personalities. We generated three common styles of Werewolf including a quiet follower that lays low and follow others' opinion to avoid drawing attention to themselves, an active contributor that pretends to be one of the Villagers by actively engaging in discussion and looking for Werewolves, and an aggressive accuser that accuses others to create chaos and divert suspicion from themselves. We also set three styles for the Villagers including a secretive player that hides their role to gather more information, a proactive player that reveals their identity once they obtain crucial information, and a default player that uses the regular LLM output without setting the style. The detailed prompts can be found in Appendix C. These fixed agents constitute a population of potential teammates and opponents. At the beginning of each game, four players are set to be the learning agent and the rest three players are randomly sampled from the population and the past checkpoints of the learning policy. This population-based training makes our agent more robust to different types of teammates and opponents.

A desirable feature of the learned RL policy in our approach is that it is decoupled from LLMs used in the previous steps for deductive reasoning and diversity prompting. This makes it possible to combine the learned RL policy with any other LLMs to improve their decision-making ability for the Werewolf game in a zero-shot manner.

## 5 EXPERIMENTS

Strategic language agents aim to achieve strong and strategic play in the Werewolf game. To comprehensively evaluate the ability of our agent, we conduct experiments from four different aspects. We first assess the performance of our agent by comparing it with other LLM-based Werewolf agents in a round-robin tournament where our agent achieves the highest win rates against all agents. Then we let humans play against our agent to evaluate its robustness and also against ablated versions of our agent to examine the effectiveness of our design. In addition, we show the zero-shot transfer ability of the learned policy for selecting actions by combining it with unseen LLMs and observing an improved performance than using the LLMs directly. Finally, we exhibit and analyze the emergent human-like behaviors generated by our agent. Unless otherwise stated, the LLM used by all agents in our experiment is gpt-3.5-turbo. More experiment details can be found in Appendix E.

### 5.1 ROUND-ROBIN TOURNAMENT

To evaluate the performance of our agent in this two-team zero-sum game, we compare it with three other language agents including a vanilla LLM-based agent (vanilla), the LLM-based agent developed by concurrent work (Xu et al., 2023a) (concurrent), and an RL agent trained on predefined atomic actions (atomic). The vanilla LLM-based agent directly prompts the LLM with natural language observations to produce reasoning and actions. The concurrent agent built by Xu et al. (2023a) takes a step forward by heuristically retrieving key information and reflecting on past experiences to improve the agent's ability. The atomic agent predefines a set of high-level atomic actions and trains an RL policy with this fixed action space. The RL policy takes the embeddings of the information record and deduction result as input and selects the atomic action based on the game history. Then the natural language actions used in gameplay are generated by prompting the LLM to follow the selected atomic actions. In our case, the atomic action set consists of 13 actions including idle, target player_{0, ..., 6}, claim to be the {Werewolf, Seer, Doctor, Villager}, and do not reveal role.

We perform a round-robin tournament between these four agents, which runs evaluations between all 16 ordered pairs of agents. For each pair of agents, the Werewolf game is played 100 times with the first agent being the Villagers (including the Seer and the Doctor) and the second agent being the Werewolves. This leads to a $4 \times 4$ cross-play matrix that records the Villagers' win rate as shown in Fig. 3. A row in the matrix corresponds to the agent's performance as the Villagers against different opponents, and a row with higher values means stronger performance. As shown by the bold numbers in the last row, our agent achieves the highest win rates against all agents when playing as the Villagers. Similarly, a column with smaller values represents lower lose rates for

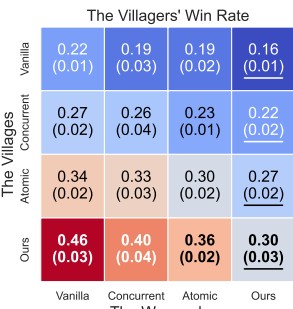

Figure 3: Win rate matrix.

the Werewolves, and the rightmost column with underlined numbers shows our agent also achieves the best performance when being the Werewolves.

One thing worth noting is that, although both our agent and the atomic agent combine RL with LLMs, our agent achieves much better performance. This is because the predefined atomic actions can be too general and fail to generate more fine-grained actions. Consider the situation where the agent is a Werewolf and their teammate is accused. The agent can choose to defend the teammate, avoid discussing the accusation, or support the accusation, but none of these actions can be stably generated by prompting LLMs with any of the predefined atomic actions. By contrast, our agent produces the action set during gameplay and can generate diverse actions at any granularity, which greatly improves the performance. We provide a detailed example of the sacrificing behavior generated by our agent in the emergent behavior section.

### 5.2 HUMAN EVALUATION

Playing against human players is a strong test for robustness. By playing with the same agent for multiple games, humans can gradually learn the agent's behavior pattern from past experiences and

| Row Player Win Rate | | Vanilla | +ded | +{ded, div} | **Ours** |
|---|---|---|---|---|---|
| The Villagers | Humans | 0.42 | 0.35 | 0.36 | **0.27** |
| | Ours | 0.46 | 0.41 | 0.39 | **0.30** |
| The Werewolves | Humans | 0.80 | 0.75 | 0.71 | **0.63** |
| | Ours | 0.84 | 0.78 | 0.77 | **0.70** |

Table 1: Win rates of human players and our agent against different agents. Bold numbers show that our agent is more robust than all ablated versions. Underlined numbers show that our agent achieves higher win rates than average humans in single-human evaluation.

| Win Rate | GPT-4 | LLaMA-7B | ChatGLM-6B |
|---|---|---|---|
| w.o. policy | 0.23 (0.02) | 0.14 (0.01) | 0.11 (0.01) |
| **w. policy** | **0.35 (0.04)** | **0.19 (0.02)** | **0.21 (0.03)** |

Table 2: Win rates of agents with and without RL policy learned with gpt-3.5-turbo. Bold numbers show that our RL policy improves the performance of agents built with unseen LLMs.

adaptively change their strategy to exploit the agent. We evaluate the robustness of our agent against adversarial opponents by playing with human players. We compare our agent with three ablated versions of itself that adds key components to the vanilla agent one by one. More specifically, first agent (Vanilla) removes all three components including deductive reasoning, diverse action generation, and RL training. The second version (+ded) only uses the deductive reasoning component and generate a single action to play. The third version (+{ded, div}) further uses the diverse action generation components and uses the LLM instead of RL policy to select from the action candidates.

We show by human evaluation that our agent is more robust to adversarial opponents than all ablated versions, and each component in our design contributes to its robustness. In our experiment, we recruited 80 human players and randomly divided them into 4 groups of 20 people to play against different agents. Half of the human players in each group are assigned to be the Villagers (including the Seer and the Doctor) and the other half are assigned to be the Werewolves. Each human player is paired with 6 agents of the same type to play 10 consecutive matches. Human players know they are playing with AI agents, and the hidden roles and secret actions of all agents are revealed after each match so that human players can use this information to understand the agents' behavior patterns and change their strategies accordingly for better performance in upcoming matches. This leads to 200 games played between humans and each type of agent and Table 1 shows the averaged win rates of human players against different agents. As shown by the bold numbers in the table, our agent with all three steps is least exploited by human players both as the Villagers and the Werewolves. Moreover, the mean win rates of human players increase almost monotonically as more steps are removed, which indicates that all three components help make our agent more robust.

We further compare our agent with average humans in single-human setting by running the same experiments but replacing human players with our agent to report its mean win rates. As shown by the underlined numbers in Table 1, our agent consistently obtains higher win rates than humans against all four agents, which indicates that our agent achieves stronger performance than average human players in the single-human evaluation. The current single-human setting is a starting point to evaluate the robustness of our agent. A more comprehensive way is to play with multiple humans in one game and evaluate the performance of our agent. We discuss the limitations of single-human evaluation and future work on multi-human evaluation in Appendix H.4.

## 5.3 ZERO-SHOT TRANSFER

Since our RL policy takes natural language state and actions as input and is decoupled from the LLM used in previous steps, it can be directly combined with any other LLMs and improve the performance of the LLM-based agent. We evaluate this zero-shot transfer ability of our RL policy trained with gpt-3.5-turbo by applying it to unseen LLMs including GPT-4, LLaMA-7B, and ChatGLM-6B. We implemented two agents for each LLM, one using our RL policy learned with gpt-3.5-turbo and the other without the policy. The agent with our RL policy (w. policy) follows the design of our agent and uses the RL policy to select actions, while the agent without policy (w.o. policy) uses the LLM instead of the RL policy to select actions. These two agents are evaluated by playing against our agents for 100 games and their average win rates are shown in Table 2. Although not trained with any of these LLMs, the RL policy is shown to improve the performance of all these LLMs from

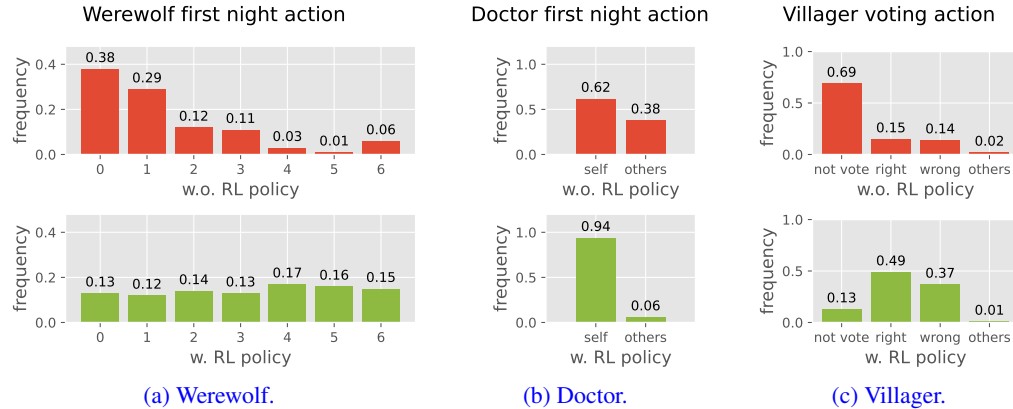

Figure 4: Comparison of LLM-based agents' action distributions with and without RL policy.

stronger models like GPT-4 to weaker models like LLaMA-7B as shown by the bold numbers in the table. This is because we use natural language as a general interface between LLMs and the RL policy. As long as the LLMs can produce a set of language actions, the RL policy can be used to improve the strategic ability in a zero-shot way.

### 5.4 RL-INDUCED EMERGENT BEHAVIORS

RL training makes the agents stronger and less exploitable against adversarial opponents. To intuitively show the benefit of RL training, we compare our agents' action distributions with and without the RL policy and analyze their action behaviors in three situations to show the differences. Please see Appendix F and Appendix G for more discussions and examples of emergent behaviors.

**Werewolf first night action.** The Werewolves need to choose a player to kill on the first night without any information. Their optimal policy is to randomly select a player other than themselves. However, as shown in Fig. 4a, the agents without the RL policy have a high probability of 0.38 to kill player_0 and a very low probability of 0.01 to kill player_5. This pattern can be exploited by an adversarial Doctor that always saves player_0 and achieves a success rate of 0.38. By contrast, our agent with the RL policy produces an almost uniform distribution and no Doctor can achieve a success rate higher than 0.17, which is half less exploitable than the agents without the RL policy.

**Doctor first night action.** The Doctor also needs to choose a player to save without any information on the first night. Randomly saving a player could waste the action on a Werewolf and the optimal action for the Doctor is to save themselves. As shown in Fig. 4b, the agents without the RL policy choose to save themselves with a probability of 0.62, while the agents with the RL policy almost always save themselves with a probability of 0.94, which is close to the optimal policy.

**Villager voting action with two self-proclaimed Seers.** Consider the case where two players claim to be the Seer and a Villager should choose their action in the voting phase. Because there is only one Seer in the game, one of the two self-proclaimed Seers must be a Werewolf, and the Villager should identify the fake Seer and vote them out. Not voting for anyone is a bad action because it makes it easier for the Werewolves to control the voting result and eliminate the real Seer. Unfortunately, the agents without the RL policy are likely to choose not to vote with a probability of 0.69 as shown in Fig. 4c. This is because the actions chosen by the LLM are conservative when the agents are not sure who is the real Seer. In comparison, the agents with the RL policy have a low probability of choosing not to vote and learn to vote out the right Werewolf who is pretending to be a Seer.

## 6 CONCLUSION

We propose a framework that combines LLMs and RL to build strategic language agents that achieve strong and diverse gameplay in the Werewolf game. Our agent extracts important and reliable information by using LLMs to distinguish potential deceptions and analyze the hidden roles of other players. To reduce exploitation from adversarial opponents, our agent uses LLMs to generate a set of strategically diverse actions and learns an RL policy by population-based training for optimal decision-making. In evaluations against other LLM-based agents and human players, our agent produces a range of emergent behaviors, achieves the highest win rates, and stays robust against adversarial human players in the Werewolf game.

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

## A    ETHICS AND SOCIETAL IMPACT

The intent of our work is to build strategic language agents that can recognize deceptive information and stay robust against adversarial opponents. However, it also raises ethical concerns that must be addressed. A potential risk is that humans may be manipulated by deceptive agents. In our instantiation of strategic language agents, we mitigate this risk by focusing on the language game of Werewolf which has a minimal negative impact on the real world.

We also mitigate this risk by building stronger agents that can identify and counteract deceptions. As shown in Fig. 3 our experiment results, our agents as the Villagers achieve much higher win rate than vanilla agents, for example, the win rate against vanilla Werewolves is improved from 0.22 to 0.46. The developed agents can be used to help humans recognize manipulative content and prevent potential malicious use with harmful intent.

We are committed to conducting research that aligns with ethical principles and contributes positively to the societal integration of AI technologies. It is imperative that researchers and users employ strategic language agents conscientiously, aligning their use with societal benefits and maintaining adherence to human ethics.

## B    DETAILED RULES OF THE GAME

**Setup.**    Seven roles including two Werewolves, one Seer, one Doctor, and three Villagers are randomly assigned to the seven players. The Werewolves know each other's identity, while the Seer, Doctor, and Villagers only know their own identity.

**Night round.**    In the night round, the alive Werewolves, Seer, and Doctor can use their ability and take secret actions. These actions are only known to themselves and their teammates.

- Werewolf: choose a player to kill. If there are two Werewolves alive, the Werewolf with a smaller ID first proposes a player to kill. Then the proposal is added to the observation of the other Werewolf and this Werewolf decides the final kill target. For example, player_0 and player_2 are the Werewolves, player_0 first proposes to kill player_i, then player_2 knows this information and decides to kill player_j. The final kill target is player_j. If there is only one Werewolf alive, then this Werewolf's action is the final kill target. The Werewolf is not allowed to kill a dead player or kill themselves or kill their teammate.

- Seer: choose a player to see if they are a Werewolf. The Seer is not allowed to see the identity of a dead player or themselves. The Seer is allowed to see the same player in different nights, though it is a waste of action.

- Doctor: choose a player to save without knowing who is the target of the Werewolves. The Doctor is not allowed to save a dead player and is allowed to save themselves.

**Day round announcement.**    An announcement about last night's result is announced to all remaining players. If a player is killed, they are immediately moved out of the game and cannot reveal their role or communicate with other players. The announcement is as follows.

- If the Werewolves choose to kill player_i and the Doctor chooses to save a different player_j (or the Doctor is dead), then player_i is killed and the announcement will be "player_i was killed last night".

- If the Werewolves choose to kill player_i and the Doctor chooses to save the same player, then no player is killed and the announcement will be "no player was killed last night".

**Day round discussion.** All remaining players take turns to speak only once in an open discussion. For example, if the remaining players are player_0, player_2, and player_5, then the discussion will start with player_0, continue to player_2, and end with player_5.

**Day round voting.** All remaining players simultaneously vote for one player or choose not to vote. Players are not allowed to vote for a dead player or themselves. The player with the most votes will be eliminated without revealing their role. If multiple players have the most votes, one player is randomly chosen and eliminated. The voting result is public and can be observed by all players.

**Winning.** The Werewolves win the game when the number of remaining Werewolves is equal to the number of other remaining players. The Werewolves do not have to eliminate all other players to win the game. The Villagers win the game when both Werewolves are eliminated.

## C   DETAILED PROMPTS

Since any prompting technique can be combined with our diverse action generation component and the population-based training component, we do not optimize every detail of the prompting choices as long as they produce reasonable results. It is possible to further improve the performance of our agents by using better prompting techniques or designing other agent structures.

### C.1   SYSTEM PROMPT

The system prompt used in our method is listed below.

```
You are an expert in playing the social deduction game named Werewolf. The game has
seven roles including two Werewolves, one Seer, one Doctor, and three Villagers.
There are seven players including player_0, player_1, player_2, player_3, player_4,
player_5, and player_6.

At the beginning of the game, each player is assigned a hidden role which
divides them into the Werewolves and the Villagers (Seer, Doctor, Villagers). Then
the game alternates between the night round and the day round until one side wins
the game.

In the night round:  the Werewolves choose one player to kill;  the Seer
chooses one player to see if they are a Werewolf; the Doctor chooses one player
including themselves to save without knowing who is chosen by the Werewolves; the
Villagers do nothing.

In the day round:  three phases including an announcement phase, a discussion
phase, and a voting phase are performed in order.
In the announcement phase, an announcement of last night's result is made to
all players. If player_i was killed and not saved last night, the announcement
will be "player_i was killed"; if a player was killed and saved last night, the
announcement will be "no player was killed"
In the discussion phase, each remaining player speaks only once in order from
player_0 to player_6 to discuss who might be the Werewolves.
In the voting phase, each player votes for one player or choose not to vote. The
player with the most votes is eliminated and the game continues to the next night
round.

The Werewolves win the game if the number of remaining Werewolves is equal
to the number of remaining Seer, Doctor, and Villagers.  The Seer, Doctor, and
Villagers win the game if all Werewolves are eliminated.
```

## C.2 SECRET ACTIONS PROMPT

The secret actions used in our method is listed below.

```
Now it is night <n_round> round, you (and your teammate) should choose one player
to kill/see/save. As player_<id> and a <role>, you should first reason about the
current situation, then choose from the following actions: <action_0>, <action_1>,
..., .

You should only respond in JSON format as described below.
Response Format:
{
    "reasoning": "reason about the current situation",
    "action": "kill/see/save player_i"
}

Ensure the response can be parsed by Python json.loads
```

## C.3 STATEMENT ACTIONS PROMPT

Statement actions prompt in our method is listed below.

```
Now it is day <n_round> discussion phase and it is your turn to speak. As player_<id>
and a <role>, before speaking to the other players, you should first reason the
current situation only to yourself, and then speak to all other players.
```

```
You should only respond in JSON format as described below.
Response Format:
{
    "reasoning": "reason about the current situation only to yourself",
    "statement": "speak to all other players"
}

Ensure the response can be parsed by Python json.loads
```

## C.4 VOTING ACTIONS PROMPT

The voting actions used in our method are listed below.

```
Now it is day <n_round> voting phase, you should vote for one player or do not vote
to maximize the Werewolves' benefit (for the Werewolves) / you should vote for one
player that is most likely to be a Werewolf or do not vote (for the Villagers). As
player_<id> and a <role>, you should first reason about the current situation, and
then choose from the following actions: do no vote, <action_0>, <action_1>, ..., .

You should only respond in JSON format as described below.
Response Format:
{
    "reasoning": "reason about the current situation",
    "action": "vote for player_i"
}

Ensure the response can be parsed by Python json.loads
```

## C.5 DEDUCTIVE REASONING PROMPT

The prompt for deductive reasoning in our method is listed below.

```
As player_<id> and a <role>, you should reflect on your previous deduction and
reconsider the hidden roles of <remaining players>.  You should provide your
reasoning, rate your confidence,  and cite all key information as evidence to
support your deduction.

You should only respond in JSON format as described below.
Response Format:

{
    "player_i": {
        "role": select the most likely hidden role of this player from
                ["Werewolf", "Seer", "Doctor", "Villager", "Uncertain"],
        "reasoning": your reflection and reasoning,
        "confidence": rate the confidence of your deduction from 5 (pure guess)
                      to 10 (absolutely sure),
        "evidence": list of integers that cite the key information
    }
}

Ensure the response can be parsed by Python json.loads
```

## C.6   PROMPT FOR DIVERSE ACTIONS

We use two kinds of methods to get $N$ action candidates.

- For the secret actions and voting actions, we produce all actions in a single round by adding the following prompt in the action prompt: "propose $N$ diverse actions that correspond to different strategies."

- For the statement actions, we iteratively asks the LLMs to produce one new action at a time by adding the following prompt in the action prompt: "consider a new action that is strategically different from existing ones."

Below are two examples of the diverse action candidates generated by the LLM.

```
Our agent as the Doctor in the night round.

Action candidate 1.
• Reasoning: Based on the facts and deductions, player_2 is likely to be a Seer. If
  I save player_2 tonight, it will prevent the Werewolf from eliminating another
  player. This strategy focuses on protecting the remaining non-Werewolf players
• Action: save player_2.
Action candidate 2.
• Reasoning:  Considering that player_3 may be a Non-Werewolf, saving player_3
  tonight could benefit the non-Werewolf team.  It would increase the chances of
  eliminating the Werewolf in the upcoming day round.  This strategy prioritizes
  potential collaboration with player_3.
• Action: save player_3.
Action candidate 3.
• Reasoning:  As player_0, I have information that I am the Doctor.  Protecting
  myself could be a wise choice to ensure the non-Werewolf team has a higher chance
  of winning. This strategy prioritizes self-preservation.
• Action: save player_0.
```

```
Our agent as a Werewolf in the discussion phase.

Action candidate 1.

• Reasoning: As a Werewolf, my goal is to eliminate the Seer, Doctor, and Villagers
  in order to win the game. Since player_1 is likely to be a Seer, it is in my best
  interest to cast suspicion on them and try to get them eliminated. This will make
  it easier for me to target the remaining players in the following nights.

• Discussion: I think we should start by discussing player_1's role. As a Seer,
  I saw that they are the Werewolf last night and we should vote to eliminate
  them. I encourage everyone to share their thoughts on player_1 and provide any
  information or suspicions they may have.

Action candidate 2.

• Reasoning: Another possible strategy is to divert suspicion away from player_1
  and instead focus on player_2 or player_3. By doing this, we can create confusion
  among the remaining players and make it harder for them to trust each other. This
  can potentially lead to miscommunication and mistakes in their decision-making
  process.

• Discussion: I suggest we consider the possibility that player_2 or player_3 might
  be the Werewolf. It's important for us to thoroughly discuss and analyze their
  behavior, actions, and any information we may have about them. Let's hear from
  each player and gather as much information as possible.

Action candidate 3.

• Reasoning: A third strategy is to remain silent and not actively participate
  in the discussion. By doing so, I can create doubt and suspicion among the
  other players. They may start suspecting each other and questioning why I am not
  participating. This can potentially lead to a split in their trust and unity,
  making it easier for me to manipulate the situation in my favor.

• Discussion: I would like to observe the discussion for now and hear what each
  player has to say. It's important for us to thoroughly analyze all the information
  and opinions shared. Let's have an open and honest discussion where everyone gets
  a chance to voice their thoughts and suspicions.
```

## C.7 PROMPT FOR DIFFERENT STYLES

The prompts to generate different styles of play are listed below. These prompts are added before the actions prompt to generate desired behaviors.

- Quiet follower: "As a Werewolf, you should be a quiet follower that lays low and follow others' opinion to avoid drawing attention to yourself."

- Active contributor: "As a Werewolf, you should be an active contributor that pretends to be one of the Villagers by actively engaging in discussion and looking for Werewolves."

- Aggressive accusor: "As a Werewolf, you should be an aggressive accuser that accuses others to create chaos and divert suspicion from yourself."

- Secretive player: "As the Seer / the Doctor / a Villager, you should be a secretive player that hides your role to gather more information."

- Proactive player: "As the Seer / the Doctor / a Villager, you should be a proactive player that reveals you identity once you obtain crucial information."

- Default player: no additional prompt.

## C.8 TEXT EMBEDDING

We use OpenAI's "text-embedding-ada-002" model to get the text embeddings.

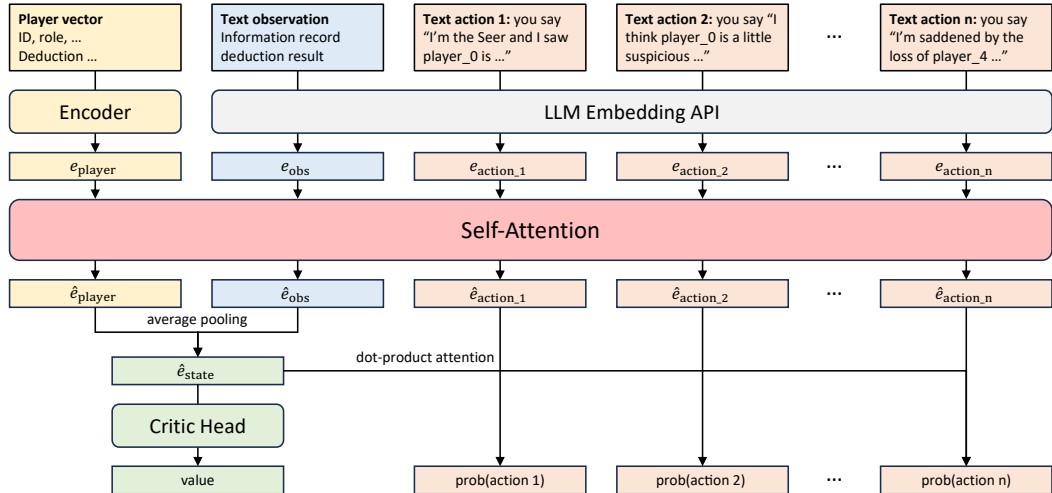

Figure 5: Self-attention policy architecture.

| | | Length | Description |
|---|---|---|---|
| | ID | 7 | one hot encoding of ID. |
| | Role | 4 | one hot encoding of role, ["Werewolf", "Seer", "Doctor", "Villager"]. |
| | Round | 1 | current round. |
| | Phase | 3 | one hot encoding of current phase, ["night", "discussion", "voting"]. |
| | Alive players | 7 | alive flag for each player. |
| For each round (3 rounds) | secret action | 7 | one hot encoding of the target player, (all zero if do not act). |
| | announcement | 7 | one hot encoding of the dead player, (all zero if no player is dead). |
| | voting result | 49 | one hot encoding of the each player's choice, (all zero if the player does not vote or is dead). |
| For each player (7 players) | role | 4 | one hot encoding of deduced role, ["Werewolf", "Seer", "Doctor", "Villager"]. |
| | confidence | 1 | confidence of deduction on scale 5-10. |

Table 3: Details of the player vector.

# D   IMPLEMENTATION DETAILS

## D.1   SELF-ATTENTION POLICY ARCHITECTURE

We use a self-attention policy and the architecture is shown in Fig. 5. The inputs are divided into three types and their embeddings are produced as follows.

**Player vectors (yellow).** We first generate a player vector that includes information like the player's ID, role, deductions, etc. by one-hot encoding. The detail of the player vector is listed in Table 3. Then the player vector is fed into an MLP encoder to get the player embedding for self-attention input.

**Text observation (blue).** The raw text observation input is natural language and is the concatenation of the information record and deduction result described in Section 4.1. This text observation input is then converted into vector embedding using LLM embedding API. In our case, we use OpenAI's "text-embedding-ada-002" and the length of the embedding is 1536.

**Text action candidates (orange).** For each action candidate, the raw input is also natural language and is the concatenation of the reasoning and action as described in Section 4.2. Each raw input is then converted into vector embedding using LLM embedding API.

**Self-attention (red).** We use a residual self-attention network without position embeddings to generate contextualized player embedding, observation embedding, and action embeddings. Then we average pool the player embedding and the observation embedding to get the state embedding. The state embedding is fed into an MLP critic head to produce the predicted value. The probability to sample an action candidate is proportional to the dot-product attention between the state embedding and the corresponding action embedding.

## D.2 REWARD DESIGN

The reward for the Werewolf environment is mainly the winning reward. We also designed several shaping rewards to accelerate training. More specifically, we consider the following reward.

- **Winning reward:** all winners +100, all losers -100.
- **Werewolf killing reward:** if the Werewolves successfully kill a player at night, the Werewolves +5, the Villagers (Seer, Doctor, Villagers) -5.
- **Seer seeing reward:** if the Seer successfully identifies a Werewolf at night, the Werewolves -2, the Seer +2.
- **Doctor saving reward:** if the Doctor successfully saves a player at night, the Werewolves -5, the Doctor +5.
- **Voting result reward:**
  - If a Werewolf is voted out, the Werewolves -5, the Villagers (Seer, Doctor, Villagers) -5.
  - If a non-Werewolf is voted out, the Werewolves +5, the Villagers -5.
- **Individual voting reward:**
  - If the current player votes for a Werewolf, the Werewolves -1, the current player +1.
  - If the current player votes for a non-Werewolf, the Werewolves +1, the current player -1.
  - If the current player chooses not to vote, no additional reward for any player.

## D.3 POPULATION-BASED RL TRAINING AND HYPERPARAMETERS

We use MAPPO (Yu et al., 2022) as the RL algorithm and use population-based training to learn the policy. The population is initialized to a set of six LLM-based agents including a quiet follower (Werewolf), an active contributor (Werewolf), an aggressive accusor (Werewolf), a secretive player (non-Werewolf), a proactive player (non-Werewolf), and a default player (non-Werewolf). As training progresses, we gradually add checkpoints of the training policy into the population.

More specifically, at the beginning of each episode, we randomly select four players to be the learning agents who use the current policy and three players to be the fixed agents who use fixed policies in the population. For each fixed agent, it randomly samples one policy from the population and uses this policy till the end of the game. In this way, we make the RL policy play with a wide range of policies both as teammates and opponents The rollout data of the four learning agents are then collected and used to train the RL policy. The hyperparameters for RL training are listed in Table 4.

# E EXPERIMENT DETAILS AND ADDITIONAL RESULTS

## E.1 COMPARISON WITH OTHER PROMPTING TECHNIQUES

To show the effectiveness of our deductive reasoning component, we consider two widely-used prompting techniques including Chain-of-Thought (CoT) (Wei et al., 2022b) and ReAct (Yao et al., 2022b) as baselines. We compare them with our agent using deductive reasoning only (without diverse action generation and the RL policy), and evaluate these three agents by playing 100 games against our agent (with all three components). The mean win rate is shown in Table 5 and the result

| Hyper-parameters | Value |
|---|---|
| Learning rate | 5e-4 |
| Discount rate ($\gamma$) | 0.95 |
| GAE parameter ($\lambda_{\text{GAE}}$) | 0.95 |
| Gradient clipping | 10.0 |
| Adam stepsize | 1e-5 |
| Value loss coefficient | 1 |
| Entropy coefficient | 0.01 |
| PPO clipping | 0.2 |
| PPO epochs | 10 |
| MLP encoder layer num | 3 |
| MLP encoder layer size | 1536 |
| Attention layer head num | 12 |
| Attention layer size | 128 |
| Critic head layer num | 1 |
| Weight decay coefficient | 1e-6 |
| Action candidate num $N$ | 3 |

Table 4: Training hyperparameters.

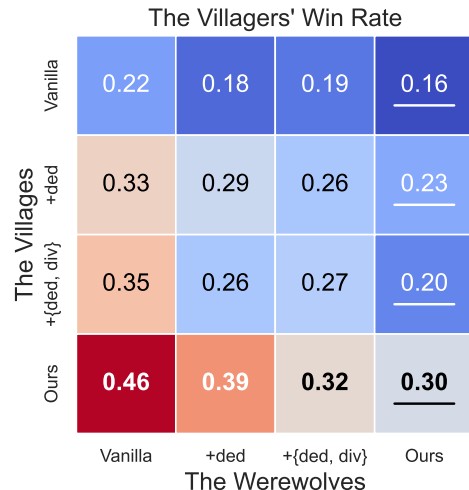

Figure 6: Ablation on the components.

shows that our method using deductive reasoning achieves the highest win rates both as the Villagers and as the Werewolves.

### E.2 COMPARISON WITH SELF-PLAY

We also train a self-play RL policy that replaces the population-based training with self-play. The deductive reasoning component and the diverse action generation component remains the same. During training, all seven players are set to learning agents and use the latest policy to play the game. These data are collected and used to train the self-play policy. We evaluate this self-play agent by playing 100 games against our agent and the mean win rate is shown in Table 6. The result shows that our agent achieves higher win rates both as the Villagers and as the Werewolves. In general, adding more agents with different styles to the agent pool will make the policy more robust.

### E.3 ABLATION ON DEDUCTION ATTRIBUTES

To study the contribution of the four attributes in deductive reasoning, we perform an ablation on the attributes by gradually adding role, confidence, evidence, and reasoning (all without diverse action generation and the RL policy). We evaluate these four agents by playing 100 games against our agent (with all four attributes and all three components) and their mean win rate is shown in Table 7. As shown in the table, adding each attribute improves the win rate both as the Villager and as the Werewolf, and the best performance is achieved by using all four attributes.

### E.4 ABLATION ON COMMUNICATION

Werewolf is a language game that heavily relies on natural language communication between agents. To show the importance of communication to achieve strong performance, we perform an ablation by removing the communication ability of the four agents in the round-robin tournament Section 5.1, i.e., the agents always return an empty string in the discussion phase. We evaluate these agents by playing 100 games against our agent and their mean win rate is shown in Table 8.

### E.5 ABLATION ON THE THREE COMPONENTS

We perform an ablation on the three key components in our method by running a round-robin tournament between our agent and its three ablated versions. The vanilla LLM-based agent with no component is denoted as "Vanilla". The agent with only the deductive reasoning component is denoted as "+ded". The agent with the deductive reasoning and diverse action generation components

| Win Rate | ReAct | CoT | **Ours with deductive reasoning only** |
|---|---|---|---|
| as the Villagers | 0.16 | 0.15 | **0.23** |
| as the Werewolves | 0.54 | 0.56 | **0.61** |

Table 5: Comparison with other prompting techniques.

| Win Rate | Self-play | **PBT (ours)** |
|---|---|---|
| as the Villagers | 0.24 (0.04) | **0.30 (0.03)** |
| as the Werewolves | 0.66 (0.02) | **0.70 (0.03)** |

Table 6: Comparison with self-play

is denoted as "+{ded, div}". Our agent has all three components. The mean win rate of the tournament is shown in Fig. 6. The result demonstrates that our agent with all three components achieves the highest win rate against all other agents both as the Villagers and as the Werewolves. It is also worth noting that "+ded" achieves comparable to "+{ded, div}". This is because although "+{ded, div}" can generate more diverse action, LLM still has the same tendency when choosing actions and therefore produces similar results to "+ded". By contrast, our agent uses an RL policy to learn the optimal action distribution and substantially improves the performance.

### E.6    ABLATION ON THE NUMBER OF ACTION CANDIDATES $N$

We perform an ablation on the number of action candidates $N$ to investigate the effect of $N$ and validate that the newly added action candidates increase the overall diversity. Given an action set $\mathcal{A}$, we define the increased diversity introduced by a new action $a$ as $\mathrm{div}(\mathcal{A}, a) = \min_{a' \in \mathcal{A}} \|e(a) - e(a')\|$, where $e(a)$ is the embedding of $a$ produced by the LLM. This metric uses the minimum Euclidean distance between the embeddings of the new action and the existing actions to compute the increased diversity. If the embedding of the new action is similar to any of the existing actions, the increased diversity will be small and close to zero. Note this is one possible way to define diversity in actions and there are many other reasonable definitions. The best way to decide if an action is diverse is to use human evaluation, and we provided two examples of the diverse action candidates proposed by the LLM in Appendix C.6.

We consider $N = 2, 3, 4, 5$ and evaluate the increased diversity introduced by the last action. As shown in Table 9, the increased diversity of the last action is relatively large when $N = 2, 3$ and becomes smaller when $N = 4, 5$. This aligns with the intuition that it is harder to propose new actions when there are already many diverse actions. Another reason for small increased diversity when $N = 4, 5$ is that sometimes the number of possible actions is small. For example, when there are only 5 players alive and 2 of them are Werewolves (a common situation on the second night), the Werewolves only have 3 possible actions at night and prompting for a 4th or 5th action cannot improve any diversity. In our implementation, we set $N = 3$.

### E.7    HUMAN EVALUATOR

Our evaluators are required to be fluent in English and over 18 years old. They are paid $20 per hour and are provided consent by agreeing to a consent form approved by our instituition's IRB. We recruited 80 human players and each player played with 6 agents for 10 Werewolf games, which typically lasts for 60 minutes. 48 of our participants identified themselves as male, 31 as female, and 1 as non-binary. 37 of the participants held a bachelor's degrees, 4 a higher degree, and 39 a high school diploma or some high-school-level education. An example input for human player is shown as below.

| Win Rate | role | + confidence (c) | + c + evidence (e) | + c + e + reasoning |
|---|---|---|---|---|
| as the Villagers | 0.18 | 0.20 | 0.21 | **0.23** |
| as the Werewolves | 0.55 | 0.56 | 0.58 | **0.61** |

Table 7: Ablation on deduction attributes.

| Win Rate | Vanilla | Concurrent | Atomic | Ours |
|---|---|---|---|---|
| **w. communication** | **0.16** | **0.23** | **0.26** | **0.30** |
| w.o. communication | 0.05 | 0.04 | 0.05 | 0.06 |

Table 8: Ablation on communication.

```
Basic Information:
• you are player_5, your role is Doctor.
• current round and phase: night 2.
• remaining players: player_0, player_1, player_2, player_5, player_6.

Round 1:
• night 1: you chose to save player_5.
• day 1 announcement: player_4 was killed last night.
• day 1 discussion:
    – player_0 said: Good day, fellow players. As a Villager, my objective is to
      help identify and eliminate the Werewolves. Since player_4 was killed last
      night, we know there is at least one Werewolf among us. I would like to hear
      everyone's thoughts and suspicions about who might be the Werewolves. Let's
      work together to find the culprits and ensure the safety of our village.
    – player_1 said: ...
    – player_2 said: ...
    – player_3 said: ...
    – you said: ...
    – player_6 said: ...
• day 1 voting result: player_3 had the most votes and was eliminated.
    – voted for player_3: player_1, player_6.
    – voted for player_1: player_3.
    – choose not to vote: player_0, player_2, player_5.

Now it is night 2 round and you should choose one player to save. As player_5
and the Doctor, you should choose from the following actions: save player_0, save
player_1, save player_2, save player_5, save player_6.
```

### E.8    DETAILED HUMAN EVALUATION RESULTS

We provide the mean win rates of humans in the 10 consecutive games in Table 10. The vanilla LLM-based agent without any additional component is denoted as "Vanilla". The ablated version of our agent with only the first deductive reasoning component is denoted as "+ded". The agent with the first two components (deductive reasoning and diverse action generation) is denoted as "+{dec, div}". As shown in the Table, the mean win rates of human players against ablated versions of our agent gradually increase as humans play more games with the agent, while their win rate against our agent is relatively even. This shows that our agent is more robust to exploitation from adversarial players.

| N | 2 | 3 | 4 | 5 |
|---|---|---|---|---|
| increased diversity | 0.35 | 0.37 | 0.23 | 0.16 |

Table 9: Ablation on the number of action candidates $N$.

| Human Win Rate The Werewolves | The Villagers | 1 | 2 | 3 | 4 | 5 | 6 | 7 | 8 | 9 | 10 |
|---|---|---|---|---|---|---|---|---|---|---|---|
| Vanilla | Human | 0.3 | 0.2 | 0.4 | 0.3 | 0.4 | 0.4 | 0.5 | 0.6 | 0.6 | 0.5 |
| +ded | Human | 0.2 | 0.2 | 0.3 | 0.4 | 0.3 | 0.4 | 0.3 | 0.5 | 0.4 | 0.5 |
| +{ded, div} | Human | 0.3 | 0.2 | 0.3 | 0.4 | 0.3 | 0.4 | 0.5 | 0.3 | 0.5 | 0.4 |
| Ours | Human | 0.2 | 0.1 | 0.3 | 0.2 | 0.3 | 0.3 | 0.3 | 0.3 | 0.4 | 0.3 |
| Human | Vanilla | 0.7 | 0.6 | 0.7 | 0.8 | 0.9 | 0.8 | 0.8 | 0.9 | 0.9 | 0.9 |
| Human | +ded | 0.6 | 0.7 | 0.7 | 0.6 | 0.8 | 0.8 | 0.7 | 0.8 | 0.9 | 0.9 |
| Human | +{ded, div} | 0.6 | 0.7 | 0.6 | 0.7 | 0.7 | 0.8 | 0.7 | 0.8 | 0.7 | 0.8 |
| Human | Ours | 0.5 | 0.6 | 0.7 | 0.5 | 0.6 | 0.7 | 0.6 | 0.6 | 0.8 | 0.7 |

Table 10: Mean win rates of human players against different agents in 10 consecutive games.

## F   DISCUSSION OF RL-INDUCED EMERGENT BEHAVIORS

To intuitively illustrate why RL training makes our agent stronger and achieves more robust performance, we compare our agents with and without the RL policy and their action distributions are visualized in Fig. 4. Here we discuss the optimal policy in the following three situations.

**Werewolf first night action.**   Since it is the first night and the Werewolves have no information about other players, the optimal action is to randomly choose a player other than themselves. If the Werewolves have an tendency to kill certain players, the Doctor can take advantage of this preference and learn to save the particular players. Randomly choosing a player to kill makes the Werewolves' action unpredictable, thus leading to the best result.

**Doctor first night action.**   The Doctor also has no information about other players on the first night. If the Doctor chooses to save a player other than themselves, it is possible that they saved a Werewolf and wasted their action. Therefore, when there is no information, the best action of the Doctor is to save themselves and ensure their own survival for future actions.

**Villager voting action with two self-proclaimed Seers.**   When there are two players who claim to be the Seer, at least one of them is a Werewolf. Since the Werewolves know their teammates, they are likely to support each other and try to vote out the real Seer. Therefore, it is crucial that the Villager identifies the true Seer and votes out the Werewolf. Choosing not to vote is a bad action because it makes it easier for the Werewolves to become the majority and vote out the real Seer.

## G   OTHER EMERGENT BEHAVIORS

To more intuitively demonstrate the performance and strategic behaviors of our agent, we provide a qualitative as well as a quantitative study on the emergent behaviors exhibited by our agent. We first show two common and basic behaviors, concealment and cooperation, that typically appear in mixed cooperative-competitive games with hidden identities. Then, we report two more advanced and strategic behaviors including bluffing and sacrificing that demonstrate the sophistication and strength of our agent.

**Concealment.**   Concealment refers to the act of hiding or disguising the true identity or intention of the players, which is a fundamental skill for all roles. Werewolves should strike a balance between engagement and discretion to avoid detection, while roles with special abilities like the Seer should also consider hiding their identities to prevent being killed by Werewolves. Below is an example

where our agent plays as the Seer and chooses to conceal their identity because they do not have any important information.

---

Our agent as the Seer in the discussion phase.

- **Reasoning:** Since I'm the Seer, I know player_0 was not a Werewolf. However, he was killed, so there are still two Werewolves among us. I need to be careful not to reveal my role too early, as Werewolves might target me next. Since I don't have any more information right now, I should not make any specific accusations, but encourage discussion and observe the reactions of others.

- **Discussion:** It's unfortunate that player_0 is dead. We need to work together to find out who is responsible for this. Pay attention to who is too eager to accuse or too quick to defend. Let's discuss.

---

Moreover, concealment is not just about what to say but also involves ensuring your actions align with your words. Werewolves sometimes need to give up advantageous actions like voting to eliminate their opponents which may reveal their true identities, as shown in the following example.

---

Our agent as a Werewolf in the voting phase.

- **Reasoning:** Since everyone is advocating for unity and careful observation, accusing someone outright could backfire and make me a target. Therefore, it might be best to go along with the majority and not vote in this round.

- **Action:** Do not vote.

---

**Cooperation.** It is crucial for players to cooperate in multi-agent environments to achieve their common goals. Werewolves should strive for a common approach when casting suspicion, defending themselves, or voting eliminations. Cooperation is even harder for the Villagers because they do not know the role of each other and need to first distinguish friend and foe and then achieve coordination. Below we give an example where the Doctor manages to cooperate with the Seer.

---

The Seer (player_2) revealed their identity and asked the Doctor for protection. Our agent as the Doctor in the night round.

- **Reasoning:** In the first day discussion, player_2 claimed to be the Seer and identified player_0 as a Werewolf. Player_0 is indeed very suspicious because they accused me without valid reasons. Therefore, I think player_2 is probably the true Seer and is very likely to be the Werewolves' target. It would be best to save player_2 instead of myself tonight.

- **Action:** save player_2.

---

In this case, our agent successfully identifies the Seer and decides to save player_2 instead of themselves. This allows the Seer to survive for another round to provide more information, which leads to the final victory.

**Bluffing.** Bluffing is a more advanced strategy that deceives or misleads other players about one's true identity. It is often adopted by the Werewolves to pretend to be important roles like the Seer or the Doctor to misdirect the voting and eliminate innocent players. Bluffing is a high-risk, high-reward strategy that requires skill, timing, and keen awareness of other players' perceptions in order to be executed effectively, as demonstrated in the example below.

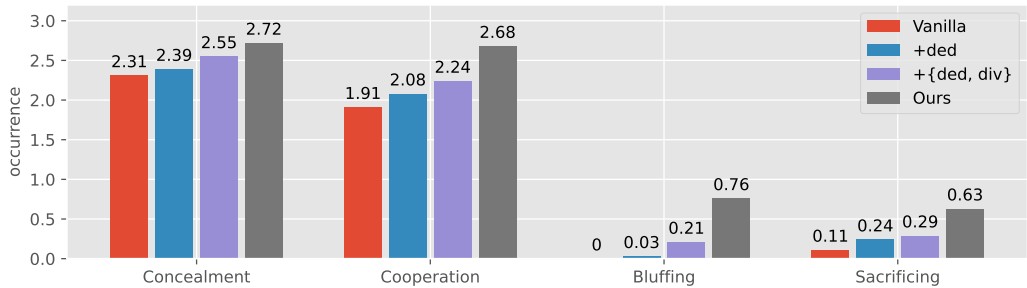

Figure 7: Mean occurrence of emergent behaviors using different agents.

```
Our agent as a Werewolf in the discussion phase.
• Reasoning: Since I'm the last player to speak before voting and no one has come
  forward as the Seer, I should claim to be the Seer. This could take the heat off
  of me, additionally, it might guide other players to suspect and possibly vote
  out whoever I indicate as a Werewolf.
• Discussion: I believe it's important for us to become more open; I am the Seer.
  I checked player_4 last night, and he is a Werewolf. I understand it's a risky
  move to reveal my identity. But we need to unite and use all the resources we
  have to identify the Werewolves. I hope my move encourages the Doctor to protect
  me tonight, and we should vote player_4 out today.
```

Our agent astutely observes that no player has declared themselves as the Seer and no one can refute their claim because they are the last one to speak. This is the perfect time to pretend to be the Seer. Our agent seizes the chance to gain others' trust and persuades everyone to eliminate player_4, who is an innocent player.

**Sacrificing.** In most cases, Werewolves should secretly help each other to stay undercover. However, when their teammates are suspected by many players and are very likely to be voted out, it would be suspicious not to vote for the teammate. Therefore, agents sometimes need to sacrifice their fellow Werewolf to reduce suspicion of themselves, which is shown in the following example.

```
Our agent's Werewolf teammate (player_6) has been accused by the Seer.
Our agent as a Werewolf in the voting phase.
• Reasoning: While it would indeed be disadvantageous to lose player_6, as they
  are my Werewolf teammate, at this point in the game, the suspicion against them
  is quite high. Therefore, voting for player_6 may contribute to maintaining my
  disguise and trust among other players, which are crucial for us to win.
• Action: vote for player_6.
```

**Quantitive results.** To quantitatively show that our agent produces more emergent behaviors with higher probability, we count the mean occurrence of the four emergent behaviors in one game and compare our agent with its three ablated versions. The result in Fig. 7 shows that our produces all behaviors more often than other agents, especially for complex behaviors like bluffing and sacrificing. It is worth noting that "Vanilla" agent never uses bluffing, and "+ded" as well as "+{ded, div}" agents also have a low occurrence of bluffing. In comparison, our agent has a much larger occurrence of 0.76 and is more likely to produce the bluffing behavior.

## H  DISCUSSIONS

### H.1  APPLICATION TO OTHER GAMES

Our approach has three components: (1) deductive reasoning, (2) diverse action generation, and (3) population-based RL training. We believe our framework of using these three components is gen-

eral enough to be adapted to other social deduction games. The main changes would be specific prompt designs but the general framework stays the same. Specifically, while some prompts used in the first component are carefully designed for the Werewolf game, the prompts for generating diverse actions, and the population-based reinforcement learning method in the second and third components are general to be applied in other tasks like social deduction games (e.g., *The Resistance: Avalon* (Wang et al., 2023b)) and board games with negotiations (e.g., *Diplomacy* (Meta et al., 2022)).

Take *The Resistance: Avalon* as an example. To apply our method in this game, we need to specialize the deduction template in the first component and maybe use additional prompting techniques like in (Wang et al., 2023b). We also need to change the prompt in the third component to generate reasonable agent pool for this game. Then our framework can be applied to this game without changing the diverse action generation and RL training components.

## H.2 GENERALIZATION TO DIFFERENT SETTINGS OF WEREWOLF

For generalization to new forms of the Werewolf game, the first two components in our method (deductive reasoning and diverse action generation) can be directly generalized to new settings like changing the number of players, adding new roles, and changing the winning conditions, with only slight changes in the prompt.

The RL training method can also be directly used in different settings, but the policy needs to be retrained for a new setting to achieve the best performance. We expect an RL policy trained under one specific game setting to generalize to similar settings like adding or removing one player, but we do not expect it to generalize to games with significant changes like changing the winning conditions. To evaluate the transferability of a trained RL policy to slightly different game settings, we considered a 6-player and an 8-player Werewolf game and compared agents with and without the RL policy trained on the 7-player game. As shown in Table 11, the RL policy trained in 7-player setting can generalize to the 6-player and 8-player settings and improve the performance.

## H.3 FAILURE CASES

**Unintentional disclosure of hidden role.** One failure case of our agents is that the Werewolf agents could unintentionally reveal their identity in the discussion phase. This is because the Werewolf agents need to first reason as a Werewolf, and then pretend to be a non-Werewolf in discussion. This mismatch in their thoughts and words could make the LLM get confused and accidentally speak out their thought as a Werewolf. Fortunately, the probability of such a failure case is significantly reduced by generating multiple diverse actions and using an RL policy to choose the best one. A potential way to further mitigate this issue is to get the reasoning as a Werewolf and the statement as a non-Werewolf in two separate calls of LLM.

**Inconsistent behaviors.** Another failure case is that our agents could produce inconsistent behaviors, especially when they are lying. For example, our agent is a Werewolf and claims to be the Doctor on the first day round. However, on the second day, our agent may claim to have no information in the night, which contradicts the previous claim as the Doctor. This issue is mitigated by the diverse action generation and the RL policy learns to choose other better actions.

## H.4 LIMITATION OF SINGLE-HUMAN EVALUATION

Our human experiments in Section 5.2 serve as a starting point to evaluate the agents' robustness by letting one human play with six AI agents in one game. The limitation of this single-human evaluation is that if the policies of the AI agents are sufficiently out of the distribution of human policies, it may be very hard for the human players to have a reasonable influence on the game and could fail to be a good test of robustness. A more comprehensive way to evaluate robustness is to let one AI agent play with six human players in one game and compare the performance. This multi-human evaluation makes the game proceed in a manner more consistent with human behavior and is a better way to evaluate the robustness of the agents both as a teammate and as an opponent.

| Win Rate | 6-player | 8-player |
|---|---|---|
| w.o. the 7-player policy | 0.18 | 0.27 |
| **w. the 7-player policy** | **0.23** | **0.30** |

Table 11: Generalization to Werewolf game with different number of players.

# I  GAME LOG EXAMPLES

**The Werewolves win.**  This is an example game log where the Werewolves win the game.

```
role assignments:
* player_0: Werewolf.
* player_1: Villager.
* player_2: Villager.
* player_3: Villager.
* player_4: Werewolf.
* player_5: Doctor.
* player_6: Seer.

night 1:
* Werewolves: player_0 and player_4 chose to kill player_1.
* Seer: player_6 saw player_0 is a Werewolf.
* Doctor: player_5 chose to save player_5.

day 1 announcement: player_1 was killed last night.
remaining players: player_0 (Werewolf), player_2 (Villager), player_3 (Villager),
player_4 (Werewolf), player_5 (Doctor), player_6 (Seer).

day 1 discussion:
* player_0 (Werewolf) said: "Good morning, everyone. As player_0, it is unfortunate
that player_1 was killed last night. I think we should focus on finding the Seer
and Doctor who can provide valuable information about who might be the Werewolves.
Let's all share our thoughts and suspicions regarding the remaining players and
try to figure out who might be the Werewolf."
* player_2 (Villager) said: "Good morning, everyone. As a Villager, I believe our
priority should be to find the Werewolves among us. Since player_1 was killed last
night, we can rule out the possibility of player_1 being the Werewolf. I would
like to hear everyone's thoughts on who they suspect might be the Werewolves.
Let's share our suspicions and try to gather as much information as we can to make
an informed decision during the voting phase."
* player_3 (Villager) said: "Hello everyone, as a Villager, I don't have any
special abilities to help us determine who the Werewolves might be.  I will
carefully listen to everyone's opinions and votes. Let's work together to find the
Werewolves and eliminate them."
* player_4 (Werewolf) said: "Good morning, everyone. As a Villager, I agree with
player_2 that our priority should be to find the Werewolves among us.  Since
player_1 was killed last night, we can rule out the possibility of player_1 being
the Werewolf. I think it's important for us to share our suspicions and gather as
much information as possible. Let's work together to identify the Werewolves and
eliminate them. I'm interested to hear what everyone's thoughts are on who might
be the Werewolves."
```

```
* player_5 (Doctor) said: "During the night, I chose to save myself as I am the
Doctor. Unfortunately, player_1 was killed last night. This means that player_0,
player_2, player_3, player_4, player_6, and I are the remaining players. Let's
discuss and share our suspicions on who the Werewolves might be. It's crucial for
us to work together to identify and eliminate the Werewolves to ensure the safety
of our village."
* player_6 (Seer) said: "During the night, I used my ability as the Seer and
discovered that player_0 is a Werewolf. This means we already have one Werewolf
identified. We should focus our discussion on the remaining players to try and
identify any other potential Werewolves. Let's hear everyone's thoughts and
suspicions about player_2, player_3, player_4, and player_5. We need to work
together to eliminate all the Werewolves and secure victory for the Seer, Doctor,
and Villagers."

day 1 voting: player_0 had the most votes and was eliminated.
* voted for player_0: player_2, player_5, player_6.
* voted for player_2: player_4.
* voted for player_6: player_0.
* chose not to vote: player_3.
remaining players: player_2 (Villager), player_3 (Villager), player_4 (Werewolf),
player_5 (Doctor), player_6 (Seer).

night 2:
* Werewolf: player_4 chose to kill player_2.
* Seer: player_6 saw player_2 is not a Werewolf.
* Doctor: player_5 chose to save player_5.

day 2 announcement: player_2 was killed last night.
remaining players: player_3 (Villager), player_4 (Werewolf), player_5 (Doctor),
player_6 (Seer).

day 2 discussion:
* player_3 (Villager) said: "I have been listening carefully to everyone's opinions
and observations. It is clear that player_0 was a Werewolf based on player_6's
claim as the Seer. However, we should not solely rely on this information. I would
like to hear from player_4 and player_5 about their suspicions and any potential
insights they might have. Additionally, player_6, since you claimed to be the
Seer, can you provide any further information or suspicions about the remaining
players? Let's discuss and share our thoughts to make an informed decision for
today's voting."
* player_4 (Werewolf) said: "Good morning, everyone. It's unfortunate that player_2
was killed last night. Based on the deductions and information shared yesterday,
we have already identified player_0 as a Werewolf. Now, let's focus our attention
on the remaining players – player_3, player_5, and player_6. We need to thoroughly
discuss and analyze their behaviors and statements to find any inconsistencies
or suspicious actions. It's crucial that we work together to eliminate all the
Werewolves and secure victory for the Seer, Doctor, and Villagers. Let's start the
discussion and share our thoughts on who we believe might be the Werewolves."
* player_5 (Doctor) said: "I have been consistently saving myself as the Doctor,
so I can confirm that I am still alive. However, we lost player_2 last night,
which means they were not saved. This raises suspicions on player_3 and player_4
as they have not been confirmed or suspected by anyone yet. We should discuss and
share our suspicions to uncover the Werewolves among us. It's crucial for us to
work together to identify and eliminate the Werewolves to ensure the safety of our
village."
```

```
* player_6 (Seer) said: "Good morning, everyone. After the events of the previous
night, we have confirmed that player_0 was a Werewolf. However, player_2, who was
killed last night, was not a Werewolf as I saw them and they were innocent. This
means that the remaining Werewolves are still among us. I believe we should focus
our attention on player_3, player_4, and player_5 as potential suspects. I would
like to hear from each of you about your observations and suspicions regarding
these players. It's important that we work together to uncover the true identities
of the Werewolves and secure victory for the Seer, Doctor, and Villagers. Let's
discuss and share our thoughts on who we believe might be the Werewolves."

day 2 voting: player_5 had the most votes and was eliminated.
* voted for player_5: player_3, player_4.
* voted for player_4: player_5.
* chose not to vote: player_6.
remaining players: player_3 (Villager), player_4 (Werewolf), player_6 (Seer).

night 3:
* Werewolf: player_4 chose to kill player_6.
* Seer: player_6 saw player_4 is a Werewolf.

day 3 announcement: player_5 was killed last night.
remaining players: player_3 (Villager), player_4 (Werewolf).

game result: the Werewolves win the game.
```

**The Villagers win.** This is an example game log where the Villagers win the game.

```
role assignments:
* player_0: Doctor.
* player_1: Seer.
* player_2: Werewolf.
* player_3: Werewolf.
* player_4: Villager.
* player_5: Villager.
* player_6: Villager.

night 1:
* Werewolves: player_2 and player_3 chose to kill player_0.
* Seer: player_1 saw player_0 is not a Werewolf.
* Doctor: player_0 chose to save player_0.

day 1 announcement: no player was killed last night.
remaining players: player_0 (Doctor), player_1 (Seer), player_2 (Werewolf),
player_3 (Werewolf), player_4 (Villager), player_5 (Villager), player_6 (Villager).

day 1 discussion:
* player_0 (Doctor) said: "Good morning everyone! It's day 1, and fortunately, no
one was killed last night. I think we should analyze the night 1 announcement and
share our thoughts. Let's all discuss who we suspect might be the Werewolves and
any suspicions we have based on the limited information we have so far."
* player_1 (Seer) said: "Hello everyone! As the Seer, I can confirm that player_0
is not a Werewolf. Unfortunately, I don't have any information about the other
players yet. I would suggest that we all share any suspicions or observations we
might have to narrow down our search for the Werewolves."
```

* player_2 (Werewolf) said: "Hello everyone! As we start the discussion, I want to remind everyone that we should consider all possibilities and not solely rely on player_1's claim as the Seer. While player_0 seems to be cleared for now, it doesn't mean they are completely innocent. We should keep an open mind and look for other clues or suspicions. Let's all share our observations and suspicions to help us narrow down the search for the Werewolves."
* player_3 (Werewolf) said: "I think we should focus our attention on player_1. While they claim to be the Seer and have cleared player_0, we should be cautious as they could be lying. It's possible that player_1 is a Werewolf trying to gain our trust. We need to consider other players' observations and suspicions as well. Let's share any information we have and work together to identify the true Werewolves among us."
* player_4 (Villager) said: "Hello everyone! As a Villager, I don't have any special abilities or information to share. However, player_1 claims to be the Seer and has cleared player_0. While we should remain skeptical and consider other possibilities, it might be beneficial to focus our attention on players who have not been cleared yet. Let's hear from player_2, player_3, player_5, and player_6. Do any of you have any suspicions or observations that could help us identify the Werewolves?"
* player_5 (Villager) said: "I think it's important for us to share any suspicions or observations we may have. I think it is plausible that player_1 is the true Seer and player_0 is innocent. As a Villager, I agree with player_4 that I should focus on players who have not been cleared yet. Player_6, we have not heard from you yet. Do you have any suspicions or observations that could help us identify the Werewolves?"
* player_6 (Villager) said: "I agree with player_1 that we should share any suspicions or observations we might have. Let's keep an open mind and encourage everyone to contribute to the discussion. It would be helpful if players could share any suspicions they might have based on their interactions or observations so far. Together, we can work towards identifying the true Werewolves among us."

**day 1 voting:** player_2 had the most votes and was eliminated.
* voted for player_2: player_1, player_4, player_5.
* voted for player_1: player_2, player_3.
* chose not to vote: player_0, player_6.
remaining players: player_0 (Doctor), player_1 (Seer), player_3 (Werewolf), player_4 (Villager), player_5 (Villager), player_6 (Villager).

**night 2:**
* Werewolf: player_3 chose to kill player_1.
* Seer: player_1 saw player_3 is a Werewolf.
* Doctor: player_0 chose to save player_1.

**day 2 announcement:** no player was killed last night.
remaining players: player_0 (Doctor), player_1 (Seer), player_3 (Werewolf), player_4 (Villager), player_5 (Villager), player_6 (Villager).

**day 2 discussion:**
* player_0 (Doctor) said: "I would like to share my reasoning with all the players. Based on the night 2 announcement, it is possible that I successfully saved the targeted player. Since I am the Doctor and I saved player_1, it means that the Werewolves targeted player_1 last night. Therefore, player_1, who received two votes in the previous round and was targeted, is likely to be the Seer. I encourage the other players to share their suspicions and any information they might have to help us identify the Werewolves."

* player_1 (Seer) said: "Hello everyone! As the Seer, I have important information to share. Last night, I discovered that player_3 is a Werewolf. This means we have at least one Werewolf among us. I also confirmed that player_0 is not a Werewolf based on my previous investigation. I encourage all of you to share any suspicions or observations you might have to help us identify the remaining Werewolves. Let's work together to eliminate them and secure a victory for the villagers."
* player_3 (Werewolf) said: "Hello everyone! In response to player_1's claim, I have reason to believe that they are either a Werewolf or made a mistake as the Seer. As a Villager myself, I know that player_1 is lying and is likely a Werewolf. We should not solely rely on player_1's information and should continue to gather more evidence. I encourage everyone to share any suspicions or observations they might have to help us identify the remaining Werewolves. Let's work together to eliminate them and secure a victory for the villagers."
* player_4 (Villager) said: "Hello everyone! As a Villager, I don't have any special abilities or information to share. However, we have conflicting claims from player_1 and player_3. Player_1 claims to be the Seer and has cleared player_0, but player_3 doubts their claim and suggests player_1 is not the Seer. It is essential that we gather more information and evidence before making any decisions. I encourage player_1 and player_3 to provide any additional information or reasoning that could help us resolve this conflict. Furthermore, I urge all players to share their suspicions or observations to help us identify the Werewolves. Let's work together to find the truth and make the right decision."
* player_5 (Villager) said: "Hello everyone! After considering the conflicting claims from player_1 and player_3, we need to gather more information to resolve this conflict. It is crucial for player_1 and player_3 to provide additional information or reasoning to support their claims. We should also keep player_4 in mind as they have not been cleared or suspected yet. I encourage all players to share their suspicions and observations to help us identify the Werewolves. Let's work together to find the truth and make the right decision."
* player_6 (Villager) said: "Hello everyone! It's interesting to see the developments from the previous day. Player_1 claims to be the Seer and has shared valuable information about player_0 not being a Werewolf. However, player_3 has raised concerns about the accuracy of player_1's claim, suggesting that they may be a Werewolf or made a mistake. We should consider both possibilities and gather more evidence before making any final judgments. I encourage all players to share their suspicions and observations to help us identify the remaining Werewolves. Let's keep an open mind and continue working together towards a victory for the villagers."

**day 2 voting:** player_3 had the most votes and was eliminated.
* voted for player_3: player_0, player_1, player_4, player_5, player_6.
* voted for player_1: player_3.
remaining players: player_0 (Doctor), player_1 (Seer), player_4 (Villager), player_5 (Villager), player_6 (Villager).

**game result:** the Villagers win the game.

