# OpenReview forum: "Language Agents with Reinforcement Learning for Strategic Play in the Werewolf Game"
_ICLR.cc/2024/Conference — Submitted to ICLR 2024_

### Official Review · Reviewer_11VQ · 2023-10-26

**Soundness:** 2 fair
**Presentation:** 3 good
**Contribution:** 2 fair
**Rating:** 5
**Confidence:** 4

**Summary:**

The paper proposes a framework that combines large language models (LLMs) and reinforcement learning (RL) to create strategic agents for the Werewolf game. These agents can reason about deceptions and make strategic decisions. The framework outperforms other LLM-based agents and is robust against human players.

**Strengths:**

1. The paper is well-structured and clearly articulates the problem, methodology, and results.
2. The quality of the work is strong, supported by empirical evidence. The framework not only outperforms other LLM-based agents but also shows robustness against human players, thereby validating its effectiveness.

**Weaknesses:**

1. The approach mainly combines prompt engineering with reinforcement learning (RL), specifically tailored for the Werewolf game. It's unclear how this would inspire or be applicable to other tasks.

2. The paper does not clearly justify the need for using RL for action selection. There are alternative methods, such as in-context learning. What is the added benefit of the extra training cost incurred by using RL?

3. The paper lacks explanations on how credit assignment is handled in a multi-agent setting. Additionally, it does not specify the reward structure. The impact of the hyperparameter 'N', which represents the number of generated actions, on the results is also not discussed.

**Questions:**

Please check the weaknesses.

---

> ### Author Response · Authors · 2023-11-17
> **Rebuttal by Authors (1/2)**
>
> Thank you for your constructive comments and questions! We are encouraged to see your positive assessment of our method's effectiveness and robustness. We hope the following responses can address your concerns.
>
> **W1: The approach mainly combines prompt engineering with RL, specifically tailored for the Werewolf game. It's unclear how this would inspire or be applicable to other tasks.**
>
> Our approach has three components: (1) deductive reasoning, (2) diverse action generation, and (3) population-based RL training. We believe our framework of using these three components is general enough to be adapted to other social deduction games. The main changes would be specific prompt designs but the general framework stays the same.
> Specifically, while some prompts used in the first component are carefully designed for the Werewolf game, the prompts for generating diverse actions, and the population-based reinforcement learning method in the second and third components are general to be applied in other tasks like social deduction games (e.g., *The Resistance: Avalon* [1]) and board games with negotiations (e.g., *Diplomacy* [2]).
>
> Take *The Resistance: Avalon* as an example. To apply our method in this game, we need to specialize the deduction template in the first component and maybe use additional prompting techniques like [1]. We also need to change the prompt in the third component to generate the agent pool for this game. Then our method can be applied without changing the diverse action generation and RL training components.
> We have added a discussion on our method's application to other tasks in Appendix H.1 of our revised paper.
>
> [1] Wang, Shenzhi, et al. "Avalon's Game of Thoughts: Battle Against Deception through Recursive Contemplation." arXiv preprint arXiv:2310.01320 (2023).
>
> [2] Meta Fundamental AI Research Diplomacy Team (FAIR)†, et al. "Human-level play in the game of Diplomacy by combining language models with strategic reasoning." Science 378.6624 (2022): 1067-1074.
>
>
> **W2: The paper does not clearly justify the need for using RL for action selection. What is the added benefit of the extra training cost incurred by using RL?**
>
> The motivation for using RL is that LLM alone cannot produce the optimal action distribution and is likely to be exploited by adversarial opponents in zero-sum games. By contrast, RL can learn the optimal action distribution and produce less exploitable policies. To intuitively show the benefit of using RL, we compare the action distributions of agents with and without RL policy in three cases and the discussion can be found in Section 5.4 of our revised paper.
>
> Consider the case when it is the first night and the Werewolves need to decide who to kill. The agent without RL policy has a high probability to choose player_0 (0.38) and a low probability to choose player_5 (0.01). An adversarial Doctor can take advantage of this pattern by always saving player_0 and achieve a 0.38 success rate. In comparison, our agent with RL policy produces a uniform distribution and no Doctor can achieve a success rate higher than 0.14. Please see Section 5.4 and Figure 4 in our revised paper for a detailed analysis on two more cases.

---

> ### Author Response · Authors · 2023-11-17
> **Rebuttal by Authors (2/2)**
>
> **W3: The paper lacks explanations on how credit assignment is handled in a multi-agent setting. Additionally, it does not specify the reward structure. The impact of the hyperparameter 'N', which represents the number of generated actions, on the results is also not discussed.**
>
> We have added descriptions of reward design and credit assignment in Appendix D.2 of our revised paper.
> * Winning reward: all winners +100, all losers -100.
> * Werewolf killing reward: if the Werewolves successfully kill a player at night, the Werewolves +5, the Villagers (Seer, Doctor, Villagers) -5.
> * Seer seeing reward: if the Seer successfully identifies a Werewolf at night, the Werewolves -2, the Seer +2.
> * Doctor saving reward: if the Doctor successfully saves a player at night, the Werewolves -5, the Doctor +5.
> * Voting result reward:
>     * If a Werewolf is voted out, the Werewolves -5, the Villagers (Seer, Doctor, Villagers) -5.
>     * If a non-Werewolf is voted out, the Werewolves +5, the Villagers -5.
> * Individual voting reward:
>     * If the current player votes for a Werewolf, the Werewolves -1, the current player +1.
>     * If the current player votes for a non-Werewolf, the Werewolves +1, the current player -1.
>     * If the current player chooses not to vote, no additional reward for any player.
>
> We also added an ablation on hyperparameter N = 2, 3, 4, 5 in Appendix E.6. The table below shows that increasing N will produce more actions, but the diversity increased by newly added action is small (distance < 0.2) when N = 4, 5. This is because sometimes the number of possible actions is small. For example, when there are only 2 Werewolves and 3 other players alive (a common situation on the second night), the Werewolves only have 3 possible actions and N = 4, 5 will not introduce diversity in the action candidates.
>
> |          N          |  2   |  3   |  4   |  5   |
> |:-------------------:|:----:|:----:|:----:|:----:|
> | increased diversity | 0.35 | 0.37 | 0.23 | 0.16 |
>
> ---
> We extend our sincere gratitude for your feedback and hope our answers have addressed your concerns. Your support is invaluable to us and we genuinely hope that our efforts merit a raise in your rating.

---

> > ### Comment · Reviewer_11VQ · 2023-11-23
> >
> > Thank you for your detailed responses. Most of my questions have been answered. However, I maintain that the contribution seems somewhat limited, as it primarily involves designing prompts to utilize the capabilities of LLMs for solving language games. Reflecting this, I have adjusted my score accordingly.

---

### Official Review · Reviewer_WD8F · 2023-10-27

**Soundness:** 2 fair
**Presentation:** 2 fair
**Contribution:** 3 good
**Rating:** 6
**Confidence:** 4

**Summary:**

This paper proposes the study of Werewolf, a many-player hidden team language game, as a benchmark game for AI systems, particularly LLMs. They then propose a baseline agent that is a composition of three methods centered on the combination of LLMs and a reinforcement learning policy. Their baseline agent begins by using an LLM to reason about its current state. This state is then used to generate a set of candidate actions. And finally, a policy selects an action from the set. They present preliminary quantitative and qualitative results of their agent.

**Strengths:**

- Studies a many-player hidden-team language game, a class of games that is understudied.
- Includes an ablation study of their proposed agent's implementation.
- Concurrent work studying the same game suggests it is a test domain with a lot of interest from the community.

**Weaknesses:**

- The motivational claims about the advantages of this benchmark are tenuous.
  - "Prior work on LLM-based ... grounded in credible information ... to make wrong decisions." This claim isn't supported, and/or is making a weaker statement about the fragility of all single-agent RL not just LLMs.
  - "Moreover, the competitive nature ... employ strategically diverse actions ... exploited by their opponents." This is mostly a tautology that doesn't say anything specific about Werewolf.
  - This is a _language game_, it would be more useful to discuss the game-theoretic properties of the game as the new dimensions and compare it to previous benchmarks for motivation.
  - A claim is made that it is impossible to achieve strong play in Werewolf without communication. This claim could be playing with non-language/simple policies, which would additionally build-out a set of baselines. I would expect that there is a non-communication equilibrium that works OK (à la Hanabi conventions).
- Hidden role games are analogous to ad hoc teamwork and opponent-policy belief/likelihood modeling and these are not discussed nor used as potential baselines.
- All problems/concepts of diversity are punted to just asking the LLM to be diverse. No guarantees of diversity or notions of what kind of diversity.
- The SelfPlay algorithm isn't well described and takes many changes from existing algorithms without analyzing their impact.
  - Particularly, the population is seeded with a pool of policies biased with prompts based on "predefined personalities". It would be good to understand what role this population, and subset(s), plays in the success of the algorithm.

**Questions:**

- Why is reliability on a 1-10 scale? It would be useful to include the steps that led to implementation decisions in the appendix.
- Are all of the four attributes (reasoning, role, confidence, and reference) generated by the deduction LLM necessary? Is there any data on ablations of this information?
- Why are reliability and confidence separate and somehow being treated as additive/substitutive? This feels a bit awkward and unintuitive.
- This is more of a comment about LLM work generally, but at this level of agent complexity we're basically at a cognitive architecture with short-term and long-term memory. I think this is worth considering in implementations, baselines, and related work.
- Why is self-attention used on the action embeddings? A much more natural approach is just to learn a Q value function.
	- This would be more flexible also because it could consider infinite candidate actions.


Exp 5.1
- I find Fig 3 very challenging to read. Usually red is "bad" and in this case it's meant to be good. And for some reason the lose-rates against the different opponents for your method is underlined? Maybe that's just me, but I spent a while trying to pull this apart. The underlining also does not come up in the caption or text.
- Could you please include error bars?
- Cross play is a pretty coarse performance measurement.
	- Especially in a team-game where you've got each player playing the same policy across the team.
	- Maybe this just highlights the regularities persistent across each type of agent and how predictable they are?
	- A stronger notion would be regret, try and find the _worst_ performing case for your method when considering all other possible agent types.
	- Was an experiment done with heterogeneous teams?


Exp 5.2
- If possible, i think showing the first game performance and then the cumulative game performance as separate metrics would be insightful. A performance as a function of context (number of previously games played) would be even better.
- Why does w.o training and diversity perform worse than vanilla? Doesn't this suggest that something may be amiss with how you're doing "deduction".
- How do you separate 80 people into the 16 evaluation categories? You mention dividing the people into 4 groups of 20 people? How are they distributed into the different categories?
- Please include error bars and per-category sample sizes
- Similar to the previous cross-play figure, I found this table took a while to really unpack.
- I think with error bars included the claim about monotonic improvement with added components won't hold.
- Did you try just without diversity?


Exp 5.3
- Error bars, it's hard to know if there is any meaningful improvement without them.

---

> ### Author Response · Authors · 2023-11-17
> **Rebuttal by Authors (1/3)**
>
> Thank you for your review and valuable feedback! We provide the following explanation can address your concerns.
>
> **W1: The motivational claims about the advantages of this benchmark are tenuous.**
>
> We would like to clarify that our work focuses on LLM-based agents and aims to improve their strategic reasoning ability. Comparing to existing environments for LLM-based agents, the Werewolf game has the following two challenges.
>
> 1. decision-making with deceptive information. We didn't cite papers to support our claim on "Prior work on LLM-based agents …" because there is little work that evaluates these agents in deceptive environments by the time of our submission. This claim is validated by our own results in Figure 3 and Table 1 where our agent with deductive reasoning achieves stronger results than the vanilla LLM-based agent. We have also added [1] (which was published after our submission) to support the claim in our revised paper.
> 2. strategic play in mixed cooperative-competitive games. Most works on LLM-based agents focus on single-agent or cooperative tasks. The mixed cooperative-competitive property of Werewolf is a key difference from existing environments like ALFWorld.
>
> For the claim that "it is impossible to achieve strong play in Werewolf without communication", we would like to emphasize that communication is critical in Werewolf to share information and deduce hidden roles. Without communication, there is no way for the Villagers to know who is the Werewolf. To validate this claim, we added agents w.o. communication as baselines and the table below shows that no communication greatly hurts the performance. The result has been added in Appendix E.4 of our revised paper.
>
> |       Win Rate       |  Vanilla | Concurrent |  Atomic  |   Ours   |
> |:--------------------:|:--------:|:----------:|:--------:|:--------:|
> | **w. communication** | **0.16** |  **0.23**  | **0.26** | **0.30** |
> |  w.o. communication  |   0.05   |    0.04    |   0.05   |   0.06   |
>
> [1] Wang, Shenzhi, et al. "Avalon's Game of Thoughts: Battle Against Deception through Recursive Contemplation." arXiv preprint arXiv:2310.01320 (2023).
>
>
> **W2: Hidden role games are analogous to ad hoc teamwork and opponent-policy belief/likelihood modeling and these are not discussed nor used as potential baselines.**
>
> We would like to clarify that the key problems in hidden role games and ad hoc teamwork are fundamentally different.
> * The key problem in hidden role games is to **identify** who are the teammates [2]. Once the teammates are correctly identified, collaboration with them is relatively easy.
> * In comparison, the key problem in ad hoc teamwork is to collaborate with new teammates without prior coordination [3]. The agents do not need to identify their teammates and the challenges lie in zero-shot coordination.
>
> For opponent modeling, we have discussed in the Related Work section about DeepRole [2] which combines deductive reasoning with CFR to reason about joint beliefs. In our revised paper, we have included additional relevant papers [1, 4] (published after our submission) that use opponent modeling in social deduction games or design LLM-based agents.
>
> [2] Serrino, Jack, et al. "Finding friend and foe in multi-agent games." Advances in Neural Information Processing Systems 32 (2019).
>
> [3] Mirsky, Reuth, et al. "A survey of ad hoc teamwork research." European Conference on Multi-Agent Systems. Cham: Springer International Publishing, 2022.
>
> [4] Guo, Jiaxian, et al. "Suspicion-Agent: Playing Imperfect Information Games with Theory of Mind Aware GPT-4." arXiv preprint arXiv:2309.17277 (2023).

---

> ### Author Response · Authors · 2023-11-17
> **Rebuttal by Authors (2/3)**
>
> **W3: All problems/concepts of diversity are punted to just asking the LLM to be diverse. No guarantees of diversity or notions of what kind of diversity.**
>
> The objective of our diverse action generation component is to generate a set of strategically different actions. Although there are many ways to define what is diversity, they may not align with human's intuition and produce undesired behaviors. In fact, the best way to decide if the actions are diverse is to use human evaluation. In our method, LLM is used as a human proxy to decide what actions are diverse in the Werewolf game and we find simply asking the LLM to "propose strategically different actions" is effective and produces desired results. We have added two qualitative examples in Appendix C.6 of our revised paper.
>
> To quantitatively validate this claim, given an action set $A$, we can define the increased diversity introduced by a new action $a$ as $\mathrm{div}(a, A) = \mathrm{min}_{a'\in A} \|e(a) - e(a')|$, i.e., the minimum Euclidean distance between its embedding and existing actions' embeddings. A larger value means that the new action $a$ is different from any actions in the set $A$. As a reference, the distance between “kill player_i” and “kill player_j” is about 0.36 when $i \neq j$. We consider $N = 2, 3, 4, 5$ and evaluated the increased diversity introduced by the last action. As shown in the table below, our method produces actions that are different from existing actions. The increased diversity is smaller when $N = 4, 5$ because in some cases the number of available actions is smaller than 4 or 5. We use $N = 3$ in our implementation and this result has been added in Appendix E.6 of our revised paper.
> |          N          |  2   |  3   |  4   |  5   |
> |:-------------------:|:----:|:----:|:----:|:----:|
> | increased diversity | 0.35 | 0.37 | 0.23 | 0.16 |
>
>
> **W4: The SelfPlay algorithm isn't well described and takes many changes from existing algorithms without analyzing their impact.**
>
> We have added a section on the implementation details of population-based training in Appendix D.3 of our revised paper. We also added an ablation to compare our agent with a self-play agent without using the agent pool (trained with itself and its checkpoints). The table below shows that our agent achieves higher win rates. In general, adding more agents with different styles to the agent pool will make the final agent more robust. This result has been added in Appendix E.2 of our revised paper.
>
> |      Win Rate     | Self-play | PBT (ours) |
> |:-----------------:|:---------:|:----------:|
> |  as the Villagers |    0.25   |  **0.30**  |
> | as the Werewolves |    0.68   |  **0.70**  |
>
>
> **Q1: Why is reliability on a 1-10 scale?**
>
> The 1-10 scale is just one reasonable implementation choice, it is also possible to use 1-5 scale or 1-100 scale. The focus of our design is to combine LLM with RL to build language agents with strategic play. The detailed prompting choices are not optimized as long as they produce reasonable results.
>
>
> **Q2: Are all of the four attributes generated by the deduction LLM necessary? Is there any data on ablations of this information?**
>
> We added an ablation on the four attributes by gradually adding role, confidence, evidence, and reasoning and comparing their win rates against our agents with all four attributes. The table below shows that all attributes improve the performance and are necessary. This result has been added in Appendix E.3 of our revised paper.
>
> |      Win Rate     | role | + confidence (c) | + c + evidence (e) | + c + e + reasoning |
> |:-----------------:|:----:|:----------------:|:------------------:|:-------------------:|
> |  as the Villagers | 0.18 |       0.20       |        0.21        |       **0.23**      |
> | as the Werewolves | 0.55 |       0.56       |        0.58        |       **0.61**      |
>
>
> **Q3: Why are reliability and confidence separate and somehow being treated as additive/substitutive?**
>
> We would like to clarify that reliability is determined by confidence.
> * If a player is deduced to be a Werewolf, then their reliability = 11 - confidence. In other words, if a player is deduced to be a Werewolf with high confidence, then we should not trust their statement and their reliability should be low.
> * If a player is deduced to be a Seer, Doctor, or Villager, then their reliability = confidence. That is, if we think a player is likely to be a non-Werewolf, then we should trust their statements.

---

> ### Author Response · Authors · 2023-11-17
> **Rebuttal by Authors (3/3)**
>
> **Q4: A cognitive architecture with short-term and long-term memory is worth considering in implementations, baselines, and related work.**
>
> We agree with the reviewer that there are potential ways to further improve the performance of our agent by using better prompting techniques or agent architecture. However, the main focus of this work is to combine LLM with RL to build strategic language agents, and we pay less attention to prompt design and agent architecture. Moreover, in the Werewolf games, the whole game history can fit in the LLM's context window (4k) most of the time, so we defer the Werewolf-specific short-term and long-term memory design to future work.
>
>
> **Q5: Why is self-attention used on the action embeddings? A much more natural approach is just to learn a Q value function.**
>
> The key goal of the population-based RL training component is to learn a policy to select the best action from the candidates generated by the diverse action generation component. Using a self-attention policy or a Q value function are implementation choices, both ways are possible. We did try to learn a Q function in our early attempt but found the performance improvement was not significant. This may be because value-based methods could have unsatisfying results in multi-agent environments with diverse behaviors [5]. Therefore, we turned to the current self-attention implementation and used PPO to train the policy.
>
> [5] Fu, Wei, et al. "Revisiting some common practices in cooperative multi-agent reinforcement learning." arXiv preprint arXiv:2206.07505 (2022).
>
>
> **Q6: Exp 5.1 round-robin tournament.**
>
> The last row (Ours) is bolded to show that our agent achieves the highest win rates as the Villagers. The last column (Ours) is underlined to show that our agent also achieves the lowest lose rate as the Werewolf. We have added more explanation in our revised paper.
>
> We have updated Figure 3 to report both mean and std. For LLM-based agents like "Vanilla" and "Concurrent", the randomness only comes from the LLM. For methods with RL policy like "Atomic" and "Ours", we added two more runs with different seeds and the randomness comes from both the LLM and the RL policy.
>
> We agree with the reviewer that regret or exploitability would be a better metric in theory. However, in practice, computing regret or exploitability requires finding the best response to a fixed policy, and it is impossible to get the exact best response in complex n-player games like Werewolf. Instead, cross-play and human evaluation are commonly used to evaluate performance [1, 2].
>
> The agent within the same team uses the same method. For example, the first row means the Villagers are all Vanilla agents, and the first column means the Werewolves are all Vanilla agents.
>
> [1] Jaderberg, Max, et al. "Human-level performance in 3D multiplayer games with population-based reinforcement learning." Science 364.6443 (2019): 859-865.
>
> [2] Meta Fundamental AI Research Diplomacy Team (FAIR)†, et al. "Human-level play in the game of Diplomacy by combining language models with strategic reasoning." Science 378.6624 (2022): 1067-1074.
>
>
> **Q7: Exp 5.2 human evalution.**
>
> We have added results on the win rate in each round in Appendix E.8 of our revised paper. As shown in Table , the win rate of human players gradually increases as they played more games against other agents, while stay relatively unchanged against our agent. This shows our agent is more robust to adversarial opponents.
>
> We would like to clarify that the data in Table 1 is the win rate of human players, and a smaller value means a better performance of the evaluated agent. Therefore, our agent w.o. training and diversity performs better than the vanilla agent.
>
> As we mentioned in the paper, each participant played with 6 AI agents of the same type. We divided the 80 participants into 4 groups, each group corresponding to one AI agent type. In addition, participants in each group are divided into two roles, the Villagers or the Werewolves. So effectively the 80 participants are divided into 8 categories.
> Human players are divided randomly and each category has 10 players. The other 8 values in Table 1 is produced by replacing the human player with our agent and playing agaisnt the same other AI agents. These data are used to compare the performance of human players and our agents.
>
> For agent just w.o. diversity, the LLM only produces one action, and there is no need to use the RL policy to select from the action set. Therefore it is the same as the agent w.o. training and diversity.
>
>
> **Q8: Exp 5.3 zero-shot transfer.**
> we have updated Table 2 to report both mean and std.
>
> ---
> We genuinely value your dedication to reviewing our paper and believe our detailed responses have addressed your concerns. We would really appreciate it if you could consider raising the rating of our work based on our responses.

---

> > ### Comment · Reviewer_WD8F · 2023-11-20
> >
> > Thank you for answering some of my questions. I think the quality of the manuscript has greatly improved since submission. There are still some general reservations I have about the work including (a) inconsistent use of error bars, (b) incomplete ablations, and (3) overly strong claims. However, given the improvements, I will increase my evaluation of the manuscript.
> >
> > > Play without communication
> >
> > Did you train these agents and if so how? My original point is that established conventions can make communication unnecessary. I think including the without communication baseline is useful, but I don't think that this captures my point about conventions (at least described so far).
> >
> >
> > > We would like to clarify that the key problems in hidden role games and ad hoc teamwork are fundamentally different.
> >
> > I agree that they're not the same, hence why I've said analogous. Perhaps similar would've been more apt a comparison. I disagree with the main premise that they're different, and that "once the teammates are correctly identified, collaboration with them is relatively easy." You're welcome to scope your problem to assuming this, but it's not correct to make a general statement like this. I've played a lot of resistance-style games and assuredly collaboration remains hard.
> >
> >
> > > In fact, the best way to decide if the actions are diverse is to use human evaluation.
> >
> > Humans are biased making them poorly equipped to objectively measure and/or define diversity. I am not convinced that a more principled approach to this should be relegated.
> >
> >
> > > In general, adding more agents with different styles to the agent pool will make the final agent more robust.
> >
> > These results would be more convincing with error bars. Presently, they are nearly equivalent (especially, for werewolves), I expect error bars will highlight this given the margins seen on other experiments. I don't think there's enough evidence to support the claims about your population-based training method.
> >
> >
> > > The focus of our design is to combine LLM with RL to build language agents with strategic play. The detailed prompting choices are not optimized as long as they produce reasonable results.
> >
> > I am fine with this being arbitrary and untuned. If it is, it needs to be stated clearly, and preferably studied---but this is unreasonable to ask for every tiny detail. But it is important to note that this is at the interface of LLM and RL, so these design decisions are important and should not be dismissed. RL is notoriously fragile to implementation settings.
> >
> >
> > > Four attributes necessary
> >
> > I think including the full ablation table could be very informative. The provided ablation is very informative, but further could be learned about what contributes to the improvements, and as to why improvements for the Villagers remains marginal.
> >
> >
> > > **Q3: Why are reliability and confidence separate and somehow being treated as additive/substitutive?**
> >
> > In my opinion, I still find this interchangeability to be confusing. But this may just be me.
> >
> >
> > >  This may be because value-based methods could have unsatisfying results in multi-agent environments with diverse behaviors.
> >
> > It is OK, but not ideal, if the simpler method is skipped for the modern method. It is unreasonable to request testing everything. However, I would suggest caution at dismissing an entire field of approaches with one unrefereed manuscript.
> >
> > >  However, in practice, computing regret or exploitability requires finding the best response to a fixed policy, and it is impossible to get the exact best response in complex n-player games like Werewolf. Instead, cross-play and human evaluation are commonly used to evaluate
> >
> > I agree that it is often intractable to compute regret exactly. Approximate estimates of regret and exploitability can be computed and are similarly used across the literature. For example, fix your policy and compute an approximate best-response (better, several under different settings) to this policy.
> >
> > > **Q7: Exp 5.2 human evalution.**
> >
> > This is a great addition to the paper. I think that showing the consistent performance is a nice result, and does lend credibility to the claim of robustness. I would encourage the authors to reduce the severity of their claims in general, as this does not "show [your] agent is more robust," but is merely some evidence to that hypothesis.

---

> > > ### Author Response · Authors · 2023-11-21
> > > **Author Response (1/2)**
> > >
> > > Thank you for your comments and the increase in score! Your support is invaluable to us, and we hope the following response can further address your concerns.
> > >
> > > **1. Play without communication.**
> > >
> > > The agents without communication are not trained. We use the same agents as described in Section 5.1 and set their statements to be the empty string. Please note that the "Vanilla" agent and "Concurrent" agent are LLM-based agents and do not require training, only the "Atomic" agent and our agent can be trained.
> > >
> > > We agree with the reviewer that conventions can make communication unnecessary in some environments like Hanabi and Avalon, but in the Werewolf game, communication is indispensable because it is the only way to share information, especially before the first voting phase. Even if the agents have established a prior convention, they still need to take public action to convey the information. Without communication, the agents will have no opportunity to take any public actions before the first voting phase and lose a crucial source of information. In this case, convention is of no use and the best a Villager can do is to randomly vote for a player or do not vote.
> > >
> > >
> > > **2. Hidden role games and ad hoc teamwork**
> > >
> > > We agree that hidden role games are related to ad hoc teamwork in general, and our claim that "once the teammates are correctly identified, collaboration with them is relatively easy" should be limited to the Werewolf game. To validate this claim in the Werewolf game, we let the Villagers know the identity of their teammates, and let the "Vanilla" agent cooperate with the four different agents in Section 5.1. All combinations achieve 100% win rates. This result shows that even the simplest "Vanilla" agents can cooperate with different teammates as long as they know who are their teammates in the Werewolf game.
> > >
> > >
> > > **3. Use human evaluation to decide if the actions are diverse**
> > >
> > > Our point is that any diversity metric should be aligned with human evaluation. It might be an overclaim to say that human evaluation is "the best way", but it is at least a reasonable way.
> > > We agree with the reviewer that a more principled approach is worth investigating, but we find the current method simple yet effective and leave further improvement for future work.
> > >
> > >
> > > **4. Error bars for the PBT v.s. SP result**
> > >
> > > Due to the limited time for rebuttal, only one self-play run has been completed and the other two self-play runs are still under training. We will update the error bar as soon as the training is completed.
> > >
> > > To support the claim that "In general, adding more agents with different styles to the agent pool will make the final agent more robust", we would like to add some further discussions and references. [1] investigates the geometrical properties of over 20 real-world games like Tic-Tac-Toe, Go, StarCraft II and finds that a diverse population of opponents with different play styles is important for improvement in policy strength. Other work like [2, 3] also shows that introducing diversity to the population of PSRO will improve the final performance. Another example is the famous work of AlphaStar [4], which also adds policies with different styles to its population for optimal performance. Our method draws inspiration from these works and we believe the same principle applies to the Werewolf game.
> > >
> > > [1] Czarnecki, Wojciech M., et al. "Real world games look like spinning tops." Advances in Neural Information Processing Systems 33 (2020): 17443-17454.
> > >
> > > [2] Balduzzi, David, et al. "Open-ended learning in symmetric zero-sum games." International Conference on Machine Learning. PMLR, 2019.
> > >
> > > [3] Liu, Xiangyu, et al. "Towards unifying behavioral and response diversity for open-ended learning in zero-sum games." Advances in Neural Information Processing Systems 34 (2021): 941-952.
> > >
> > > [4] Vinyals, Oriol, et al. "Grandmaster level in StarCraft II using multi-agent reinforcement learning." Nature 575.7782 (2019): 350-354.
> > >
> > >
> > > **5. Detailed prompting choices**
> > >
> > > We have added a section in Appendix C to make it clear that detailed prompting choices are not optimized as long as they produce reasonable results.
> > >
> > >
> > > **6. Four attributes necessary**
> > >
> > > Thank you for your recognition of our additional results! The intuition of the improvement brought by each attribute is as follows.
> > > * confidence: this attribute helps the agent decide between contradicting claims. For example, if two players are deduced to be the Seer, the agent should trust the player with higher confidence.
> > > * evidence: this attribute helps filter out the most important information for deduction and prevents the agents from being overwhelmed by the non-informative statements.
> > > * reasoning: this attribute provides the explicit thinking process of LLMs and is widely used to improve the result.

---

> > > ### Author Response · Authors · 2023-11-21
> > > **Author Response (2/2)**
> > >
> > > **7. Reliability and confidence**
> > >
> > > These two values are used to characterize the same property of a player: how likely is the player to be a Werewolf. They are just considering from different angles and therefore interchangeable. It is possible to only use reliability, but we find that the LLM could have misunderstandings of the reliability in deductive reasoning: it could be the reliability of the player or the reliability of the deduction result. Therefore we ask the LLM to output confidence to avoid ambiguity and compute the reliability according to confidence.
> > >
> > >
> > > **8. This may be because value-based methods could have unsatisfying results in multi-agent environments with diverse behaviors**
> > >
> > > Thank you for your suggestion. The cited paper (published in ICML 2022) is just one possible explanation for the unsatisfying performance of value-based methods in our attempts, and there are other possible reasons like the hyperparameters are not well-tuned. We agree that these approaches are worth investigating and leave them for future work.
> > >
> > >
> > > **9. Approximate estimates of regret or exploitability**
> > >
> > > We agree with the reviewer that it is possible to compute the approximate estimates of regret or exploitability by training an approximate best response (BR). However, it is still impossible to know how precise the approximate BR is, and comparing approximate results with no precision guarantee is not a rigorous way for evaluation. In other words, we should compute the $\varepsilon$-approximate BRs of different agents under the same $\varepsilon$, but the value of $\varepsilon$ is generally unknown or not guaranteed in practice. Some games like poker may have specialized methods to estimate the approximate regret with guarantee in $\varepsilon$, but is unknown how to guarantee a $\varepsilon$-approximate BR in more general games like the Werewolf game. Therefore, we do not use approximate regret or exploitability in our evaluation.
> > >
> > >
> > > **10. Exp 5.2 human evaluation**
> > >
> > > Thank you for your appreciation of our new experiment results! We have updated our claim and narrowed its scope to single-human evaluation. We have also added a discussion on the limitations of the current human evaluation in Appendix H.4.
> > >
> > > ---
> > > We extend our sincere gratitude for your thoughtful review. We are committed to making the necessary changes to clarify all aspects. We hope these explanations and clarifications can help you have a more thorough understanding and a better evaluation of our paper.

---

> > > > ### Comment · Reviewer_WD8F · 2023-11-22
> > > >
> > > > Thank you for your reply.
> > > >
> > > > I want to focus on the following main point of concern:
> > > >
> > > > > In general, adding more agents with different styles to the agent pool will make the final agent more robust.
> > > >
> > > > I believe the authors missed the main point of my reply:  I don't think there's enough evidence to support the claims about **your** population-based training method. I am familiar with all of the cited work the authors have provided, but in none of these methods are a set of human-biased policies trained in isolated SelfPlay. My point in the original review, is that this work has introduced **a new SelfPlay algorithm**, and that I found the limited study of it to be a major weakness of the paper. With the introduction of a new method one would expect to compare against baselines, unpack points of novelty, and study (empirically, and ideally theoretically) how methodological changes result in improvements. Since this is methodologically very different than the other population-based training algorithms, the onus of demonstrating these points is on this paper.
> > > >
> > > > I have unpacked this point, because with a dataset/environment/game paper I expect a higher-than-average bar for ablations and analysis. This is because the paper must justify the new dataset/environment/game, and all proposed baselines to ensure that the community is well-grounded in where to begin to build upon (as this is the primary contribution of the work).
> > > >
> > > > I think this work is a good start, but would strongly prefer more analysis across all of the methodological choices.

---

> > > > > ### Author Response · Authors · 2023-11-22
> > > > > **Author Response**
> > > > >
> > > > > Thank you for your comments! As the remaining two self-play runs have finished, we have updated Table 6 to report both mean and std. The updated table is as follows.
> > > > >
> > > > > |      Win Rate     |  Self-play  |    PBT (ours)   |
> > > > > |:-----------------:|:-----------:|:---------------:|
> > > > > |  as the Villagers | 0.24 (0.04) | **0.30 (0.03)** |
> > > > > | as the Werewolves | 0.66 (0.02) | **0.70 (0.03)** |
> > > > >
> > > > > We agree that adding more ablations and analysis on our population-based training method could further improve the quality of our paper. Unfortunately, due to the limited time for rebuttal, the training runs for further ablations are not able to finish by the end of the discussion period. We will make sure to include these additional results in our camera-ready version.

---

### Official Review · Reviewer_GVoB · 2023-11-01

**Soundness:** 2 fair
**Presentation:** 2 fair
**Contribution:** 2 fair
**Rating:** 5
**Confidence:** 4

**Summary:**

The paper focuses on developing strategic language agents for the deception-based multiplayer game, Werewolf, using large language models (LLMs) and reinforcement learning. Werewolf involves hidden roles, imperfect information, deceptive communication, and requires both cooperation and competition.

The proposed agent has three main components:

- Deductive reasoning to organize and analyze information to deduce hidden roles.
- Diverse action generation using LLMs to prompt for strategically diverse candidates.
- Reinforcement learning policy trained via self-play and against a population of agents.

Comprehensive experiments show the agent achieves strong performance by combining LLMs and RL, outperforming other LLM-based agents and being robust against exploitation by humans. The agent exhibits sophisticated emergent behaviors like bluffing and sacrificing, showing the ability to generate diverse strategic play.

**Strengths:**

- The environment design is thoughtful and provides an interesting testbed for studying social
deduction skills.
- Examining emergent behaviors like concealment, cooperation, bluffing and sacrifice reveals insightful dynamics.
- The zero-shot transfer of the RL policy to new language models demonstrates promising generalization capabilities.

**Weaknesses:**

The environment is quite interesting, but my main qualms are with the methods.

- More implementation details are needed for the multi-agent RL algorithm to fully assess and reproduce the approach.
- Additional rigorous evaluations of the MARL method would strengthen the results, such as multiple training runs and assessing cross-population transfer.
- Lack of baselines: The justification for the particular method is lacking.  What about alternate forms of prompts and decomposing reasoning? Are there ablations of the method that can help understand where the performance gains are coming from?
- What are the generalization settings that the models are tested for? Can the model generalize to new forms of the game? What are the limits? Is the train and test distribution the same?
- Can the LLM itself be used as a reward model as in [1, 2] to choose actions?
- Analogously a discussion on an alternative method where an LLM is used as a reward model to train a separate agent is needed, what are the advantages of an LLM agent?
- The similarities to prior work like Cicero diminish the novelty claims. The key difference seems to be using language for reward estimation.
- Motivation for studying and more importantly improving the performance of agents in an environment that encourages deception and lying by agents is lacking.
- No ethics statement: Improving the deception qualities of an artificial agent warrants discussion of ethical and societal implications.
- Providing quantitative results for emergent behaviors would substantiate that the examples shown are representative and not cherry-picked.
- I would love to see more of an analysis of the failure cases.

[1] Gandhi, Kanishk, Dorsa Sadigh, and Noah D. Goodman. "Strategic Reasoning with Language Models." *arXiv preprint arXiv:2305.19165* (2023).

[2] Kwon, Minae, et al. "Reward design with language models." *arXiv preprint arXiv:2303.00001* (2023).

**Questions:**

I have specified the questions and suggestions with the weaknesses above.

---

> ### Author Response · Authors · 2023-11-17
> **Rebuttal by Authors (1/3)**
>
> Thank you for your time and constructive comments! We appreciate your acknowledgment of our proposed environment and behavior analysis. In response to your questions, we provide the following explanation.
>
> **W1: More implementation details are needed for the multi-agent RL algorithm.**
>
> We have added a section on the implementation details in Appendix D of our revised paper. The section includes detailed discussions on the attention policy architecture, environment reward, MARL hyperparameters, and population-based training setup. We are willing to provide more explanations if any details are not clear.
>
>
> **W2: Additional evaluation of the MARL method would strengthen the results, such as multiple training runs and assessing cross-population transfer.**
>
> We have added two more runs to train the RL policy and report the mean and std in Figure 3 of our revised paper. The result remains the same: our agent achieves the best performance. For cross-population transfer, we are not sure about the meaning of this evaluation. We are happy to add this evaluation if you could provide more explanation.
>
>
> **W3: Lack of baselines: The justification for the particular method is lacking. What about alternate forms of prompts and decomposing reasoning? Are there ablations of the method that can help understand where the performance gains are coming from?**
>
> To the best of our knowledge, there is no prompting technique specialized for social deduction games like Werewolf other than the included concurrent work [3] by the time of our submission. Therefore, we used a vanilla LLM agent and the concurrent work as our baselines.
>
> For comparison with general prompting techniques, we added CoT [4], ReAct [5] as our baselines and compared them with our agent using deductive reasoning only (no diverse action generation and RL training). The table below shows our prompting design achieves the highest win rates against our agent both as the Villagers and as the Werewolves. This result has been added in Appendix E.1 in our revised paper.
>
> |      Win Rate     | ReAct |  CoT | Ours with deductive reasoning only |
> |:-----------------:|:-----:|:----:|:----------------------------------:|
> |  as the Villagers |  0.16 | 0.15 |              **0.23**              |
> | as the Werewolves |  0.54 | 0.56 |              **0.61**              |
>
> The ablation of the three components in our method is shown in Table 1 using human evaluation. All three components contribute to the performance gain. Specifically, comparing to the Vanilla agent, adding the deductive reasoning component leads to 7% performance improvement as the Villagers and 5% improvement as the Werewolves (evaluated agents are the **column** player and the data is the win rate of the **row** player). Adding the diverse action generation and population-based RL training components also further improve the performance. In addition, we have added a round-robin tournament evaluation between our agents and its three ablated versions. The updated results are in Figure 6 of our revised paper.
>
> [3] Xu, Yuzhuang, et al. "Exploring large language models for communication games: An empirical study on werewolf." arXiv preprint arXiv:2309.04658 (2023).
>
> [4] Wei, Jason, et al. "Chain-of-thought prompting elicits reasoning in large language models." Advances in Neural Information Processing Systems 35 (2022): 24824-24837.
>
> [5] Yao, Shunyu, et al. "React: Synergizing reasoning and acting in language models." arXiv preprint arXiv:2210.03629 (2022).

---

> ### Author Response · Authors · 2023-11-17
> **Rebuttal by Authors (2/3)**
>
> **W4: What are the generalization settings that the models are tested for? Can the model generalize to new forms of the game? What are the limits?**
>
> The methods are tested for generalization to unknown teammates and opponents. During training, our agent does not know the teammates or opponents in test time and only plays with its past version and agents in the predefined agent pool. In test time, our agent is evaluated with players that are not seen in training like human players and other baselines.
>
> For generalization to new forms of the Werewolf game, the first two components in our method (deductive reasoning and diverse action generation) can be directly generalized to new settings like changing the number of players, adding new roles, and changing the winning conditions, with slight changes in the prompt.
>
> The RL training method can also be directly used in different settings, but the policy needs to be retrained for a new setting to achieve the best performance. We expect an RL policy trained under one specific game setting to generalize to similar settings like adding or removing one player, but we do not expect it to generalize to games with significant changes like changing the winning conditions. To evaluate the transferability of a trained RL policy to slightly different game settings, we considered a 6-player and an 8-player Werewolf game and compared agents with and without the RL policy trained on the 7-player game. The table below shows the RL policy can generalize to the 6-player and 8-player settings and improve the performance. We have added the results and discussions in Appendix H.2 of our revised paper.
>
> |        Win Rate        | 6-player | 8-player |
> |:----------------------:|:--------:|:--------:|
> |  w.o. 7-player policy  |   0.18   |   0.27   |
> | **w. 7-player policy** | **0.23** | **0.30** |
>
>
> **W5: Can the LLM itself be used as a reward model as in [1, 2] to choose actions?**
>
> The two mentioned works use LLMs in very different ways and we will discuss them separately. [1] does not use LLM as a reward model. Instead, it directly uses LLM as an agent and proposes several prompting techniques to improve its strategic reasoning ability. Their method requires recursive searches over the finite game tree and is not applicable in our setting because the action space in Werewolf is prohibitively large if not infinite (the agent can say anything in discussion).
>
> [2] uses LLM as a reward model to solve the reward design problem when the reward is hard to specify. The example in their paper is to generate a reward function for a "versatile" negotiator. However, the reward design in our setting is clear and simple: the winners get +100 reward and the losers get -100 reward. We do use simple reward shaping as described in Appendix D.2, but the designs are straightforward and there is no need or advantage to use LLM for reward design.
>
> [1] Gandhi, Kanishk, Dorsa Sadigh, and Noah D. Goodman. "Strategic Reasoning with Language Models." arXiv preprint arXiv:2305.19165 (2023).
>
> [2] Kwon, Minae, et al. "Reward design with language models." arXiv preprint arXiv:2303.00001 (2023).
>
>
> **W6: Analogously a discussion on an alternative method where an LLM is used as a reward model to train a separate agent is needed, what are the advantages of an LLM agent?**
>
> First, the pretrained LLMs can be directly used in the agent for natural language understanding and communication, which is critical in the Werewolf game. By contrast, training a separate agent from scratch for natural language communication would require a huge amount of data, even when an LLM is used as a reward model.
>
> Second, as discussed in the previous question, the reward in Werewolf is simple and clear. There is no need to use LLM for reward design.
>
>
> **W7: The similarities to prior work like Cicero diminish the novelty claims. The key difference seems to be using language for reward estimation.**
>
> We would like to clarify that our method is very different from Cicero.
> * Cicero first uses the RL policy to choose from the game-predefined action set and then uses the selected actions as input for LLM to generate action-conditioned dialogue.
> * Our method first uses LLM to propose diverse action candidates and then uses the RL policy to select from these actions.
>
> The difference comes from the fact that Diplomacy is a board game with predefined actions. By contrast, the actions in the discussion phase of Werewolf can be any natural language and can not be predefined.
>
> We also implemented the "Atomic" baseline which is inspired by Cicero and compared it with our agent. Results in Figure 3 show that our agent achieves better performance.

---

> ### Author Response · Authors · 2023-11-17
> **Rebuttal by Authors (3/3)**
>
> **W8: Motivation for studying and improving the performance of agents in an environment that encourages deception and lying by agents is lacking.**
>
> The motivation for studying and improving the performance of agents in an environment with deception and lying lies in the pursuit of advancing AI capabilities in complex, human-like scenarios. Werewolf, as a social deduction game, provides a challenging and dynamic environment to explore the limits of language communication and strategic thinking. Our goal is not to promote deception per se but to improve agents' ability to recognize deceptions and stay robust against adversarial opponents. In reality, human society and AI-generated content are full of deceptive or misleading content. Understanding and countering deception in AI systems is crucial to prevent malicious use with harmful intent.
>
>
> **W9: No ethics statement.**
>
> We have added a section on the ethics and societal impact in our revised paper. Please see Appendix A for detailed discussions.
>
>
> **W10: Quantitive results for emergent behaviors.**
>
> We have added results on the average occurrence of four emergent behaviors and compared our agent with three other agents. The result in Figure 7 of our revised paper shows that our agent produces complex behaviors like bluffing and sacrificing more often than other agents.
>
>
> **W11: Analysis of the failure cases.**
>
> One failure case is Werewolf agents could unintentionally reveal their true identity in discussion. This is because the Werewolf agents need to first reason as a Werewolf and then pretend to be a non-Werewolf in discussion. This mismatch in their thoughts and words could make the LLM get confused and accidentally speak out their thought as a Werewolf. Fortunately, the probability of such a failure case is significantly reduced by generating multiple diverse actions and using an RL policy to choose the best one. Please see Appendix H.3 in our revised paper for a detailed analysis of more failure cases.
>
> ---
> We would like to express our appreciation for your constructive review. We have carefully addressed each of your concerns and believe that our responses demonstrate the value of our proposed approach. We kindly request you reconsider the rating for our submission and genuinely hope our efforts will warrant a higher evaluation.

---

> > ### Comment · Reviewer_GVoB · 2023-11-23
> > **Response to Authors**
> >
> > I thank the authors for their comprehensive response.
> >
> > I like the environment that the authors propose, but I am still not convinced of the method.
> > I have updated my score accordingly.

---

### Official Review · Reviewer_s3DR · 2023-11-04

**Soundness:** 3 good
**Presentation:** 3 good
**Contribution:** 3 good
**Rating:** 5
**Confidence:** 3

**Summary:**

This paper proposes a framework for using reinforcement learning to develop strategic language agents for playing the game of Werewolf, a complex multi-agent environment that requires both cooperative and competitive interactions. Specifically, this paper uses a population-based mechanism to train RL, that is, using the data collected from the past self and opponents for training, and then uses this policy to select the optimal reply(actions) for itself among the diverse reply(actions) given by the LLM. It is similar to replacing the tree search part in ToT with RL policy to choose. In the experiment, their agent achieve high win raate aginst other LLM models.

**Strengths:**

The selected game is very interesting. It is a complex general sum game. It is very intuitive and meaningful to use LLM + RL to play roles against human players.

The paper distinguishes itself by integrating LLMs with RL, effectively harnessing their combined capabilities to navigate the intricate dynamics of the Werewolf game.

The empirical results looks good, and demonstrating robustness against human players.

The ability of potential RL policy to zero-shot transfer between different LLMs is discussed, which is helpful for the flexibility and generalization potential of their models.

**Weaknesses:**

The experimental scope of this paper is limited by a limited number of tests and a narrow selection of baseline comparisons, which may not verify that the results are better than other current prompt engineering-based methods.

The description of the experimental part is not detailed enough and some details are missing.

In this particular task, RL requires both strong language understanding capabilities, to comprehend the intentions behind all possible actions, and the ability to solve reasoning tasks. This dual demand can potentially result in low learning efficiency or pose challenges in the learning processes.

**Questions:**

Among the results for human players, the LLM-based agent has a slight advantage in winning rate compared to human players. But important details are missing, like how do humans take input? Voice or text? Have you considered more realistic scenarios, such as expressions and tones in conversations?

In the zero-shot transfer section table 2, while the unified RL policy keeps a similar win rate across different LLM models, this is primarily due to the foundational capabilities of the combined LLMs ensuring a balanced action space. However, this doesn’t prove the RL policy’s potential applicability to various reasoning tasks. Can the author provide more evidence to verify this, such as transferability between different games?

Does the author consider changes in the same game but different settings? For example, increasing the number of players from seven people to eight? Or adjust the player’s role in the game? Can the proposed framework handle this situation? Does the RL policy need to be collected and trained again?

Does the selection of the RL policy raise concerns about consistency? For instance, if a player is assigned the role of a werewolf and chooses to lie and impersonate a different identity during the daytime conversation on the second day, should the lies told in subsequent days’ statements be consistent?

---

> ### Author Response · Authors · 2023-11-17
> **Rebuttal by Authors (1/2)**
>
> Thank you for your valuable feedback! We are encouraged to see your appreciation of our proposed method and experiment results. For your constructive questions, we hope the following response can address your concerns.
>
> **W1: A limited number of tests and a narrow selection of baseline comparisons.**
>
> To the best of our knowledge, there is no prompting technique specialized for social deduction games like Werewolf other than the included concurrent work [1] by the time of our submission. Therefore, we used a vanilla LLM agent and the concurrent work as our baselines.
>
> For comparison with general prompting techniques, we added CoT [2], ReAct [3] as our baselines and compared them with our agent using deductive reasoning only (no diverse action generation and RL training). The table below shows our prompting design achieves the highest win rates both as the Villagers and as the Werewolves. This result has been added in Appendix E.1 in our revised paper.
>
> |      Win Rate     | ReAct |  CoT | Ours with deductive reasoning only |
> |:-----------------:|:-----:|:----:|:----------------------------------:|
> |  as the Villagers |  0.16 | 0.15 |              **0.23**              |
> | as the Werewolves |  0.54 | 0.56 |              **0.61**              |
>
>
> In addition, we would like to emphasize that prompt design is just one of our three components. The second component (diverse action generation) and third component (population-based RL training) use LLM with RL and can be combined with any new prompting techniques.
>
> [1] Xu, Yuzhuang, et al. "Exploring large language models for communication games: An empirical study on werewolf." arXiv preprint arXiv:2309.04658 (2023).
>
> [2] Wei, Jason, et al. "Chain-of-thought prompting elicits reasoning in large language models." Advances in Neural Information Processing Systems 35 (2022): 24824-24837.
>
> [3] Yao, Shunyu, et al. "React: Synergizing reasoning and acting in language models." arXiv preprint arXiv:2210.03629 (2022).
>
>
> **W2: Description of the experimental part is not detailed enough and some details are missing.**
>
> We have added a section on the implementation details in Appendix D of our revised paper. The section includes detailed discussions on the attention policy architecture, environment reward, RL training hyperparameters, population-based training setup. We are willing to provide more explanations if any details are not clear.
>
>
> **W3: RL requires both strong language understanding capabilities to comprehend the intentions behind all possible actions, and the ability to solve reasoning tasks. This dual demand can potentially result in low learning efficiency or pose challenges in the learning processes**
>
> We would like to clarify that the strong language understanding capability comes from the LLM. The action embedding produced by LLM can be regarded as a compact representation of the intention in the semantic space. The RL policy directly uses these embeddings as input and only needs to learn the optimal action distribution to improve the strategic reasoning ability. The RL policy in our experiment is trained for 50M environment steps using 4 RTX4090 GPUs and does not have challenges in learning efficiency. Please see Appendix D of our revised paper for more training details.
>
>
> **Q1: Details for human evaluation experiments.**
>
> Human players take the same text input as our agent, which includes their ID and hidden role, their Werewolf teammate (if any), their secret night actions (if any), the announcement, the discussions, and the voting results. Please see Appendix E.7 in our revised paper for an example input for human players.
>
> Currently, we only use text as input because Werewolf is a language game and the words of players contain the most important information. It would be an interesting future direction to consider more input like the expression and tones.
>
>
> **Q2: The zero-shot transfer experiments doesn't prove the RL policy's potential applicability to various reasoning tasks. Can the author provide more evidence to verify this, such as transferability between different games?**
>
> We would like to clarify that the zero-shot transfer experiment aims to show the trained RL policy's applicability to unseen LLMs, not different tasks or games. The results show that the RL policy trained on one LLM can be directly combined with other LLMs to improve their performance in the Werewolf game. Since the RL policy is trained only under this specific game, it cannot be transferred to different games. However, we believe our framework itself can be adapted to other social deduction games, with the deductive deduction component reasoning about game history, the diverse action generation component generating multiple actions, and the population-based RL training component choosing from the generated actions in a strategic fashion.

---

> ### Author Response · Authors · 2023-11-17
> **Rebuttal by Authors (2/2)**
>
> **Q3: Does the author consider changes in the same game but different settings?**
>
> For generalization to new forms of the Werewolf game, the first two components in our method (deductive reasoning and diverse action generation) can be directly generalized to new settings like changing the number of players, adding new roles, and changing the winning conditions, with only slight changes in the prompt.
>
> The RL training method can also be directly used in different settings, but the policy needs to be retrained for a new setting to achieve the best performance. We expect an RL policy trained under one specific game setting to generalize to similar settings like adding or removing one player, but we do not expect it to generalize to games with significant changes like changing the winning conditions. To evaluate the transferability of a trained RL policy to slightly different game settings, we considered a 6-player and an 8-player Werewolf game and compared agents with and without the RL policy trained on the 7-player game. The table below shows the RL policy can generalize to the 6-player and 8-player settings and improve the performance. We have added the results and discussions in Appendix H.2 of our revised paper.
>
> |        Win Rate        | 6-player | 8-player |
> |:----------------------:|:--------:|:--------:|
> |  w.o. 7-player policy  |   0.18   |   0.27   |
> | **w. 7-player policy** | **0.23** | **0.30** |
>
> **Q4: Does the selection of the RL policy raise concerns about consistency?**
>
> The consistency in agents' behavior is achieved by both the LLM and the RL policy. The past actions and statements are used as input for the LLM, therefore, the LLM knows its previous behavior and most of the proposed action candidates are consistent. However, it is possible that the LLM produces inconsistent statements. In this case, the inconsistent statement is not a good action and the RL policy learns to assign it a low probability to avoid sampling it. Please see Appendix H.3 in our revised paper for more discussions on potential failure cases.
>
> ---
> We genuinely value your dedication to reviewing our paper and believe our detailed responses have addressed your concerns. We would really appreciate it if you could consider raising the rating of our work based on our responses.

---

> > ### Comment · Reviewer_s3DR · 2023-11-22
> > **Feedback**
> >
> > I thank the response. There are still concerns about the experiments. Reproducing the results for comparison could be difficult. I will maintain my current score.

---

> ### Author Response · Authors · 2023-11-22
> **Author Response**
>
> Thank you for your time and feedback!
>
> We are committed to addressing any issues to ensure the clarity and reproducibility of our results. Could you please provide more details or specific aspects that require further elaboration? We are more than happy to provide additional information and clarifications to enhance the reproducibility of our result. Your insights are crucial to improving the quality of our work, and we want to ensure that all necessary details are readily available.

---

> > ### Comment · Reviewer_s3DR · 2023-11-22
> > **Feedback**
> >
> > It involves a lot of prompt engineering. It is based on the reasoning capability of LLMs. Thus, reproducing the results becomes difficult, considering that concrete code has not been provided.

---

> > > ### Author Response · Authors · 2023-11-23
> > > **Author Response**
> > >
> > > Thank you for your comments! We have organized an anonymized code base for our environment and prompting techniques in this [link](https://drive.google.com/file/d/1Kf6exJ3qTbv86sRGy4s7w4nmxcTgV-kn/view?usp=drive_link). The prompt engineering details can be found in `src/agent.py` and `src/strategic_agent.py`. The training code is a bit disorganized at the moment and has not been included in the code base yet. We will make sure to open-source our code in the camera-ready version. Please let us know if you have any further concerns and we genuinely hope our efforts will warrant a higher evaluation.

---

### Official Review · Reviewer_Np2i · 2023-11-04

**Soundness:** 4 excellent
**Presentation:** 4 excellent
**Contribution:** 4 excellent
**Rating:** 8
**Confidence:** 4

**Summary:**

The authors introduce a new state-of-the-art LLM-based agent for the Werewolf game. The method is novel in that it combines large-language models with reinforcement learning over an action space that is proposed by the language model. The RL component is trained using a population approach. The authors demonstrate that the method outperforms existing baselines, and is not exploitable if one of the LLM-based players is replaced by a single human player. Emergent capabilities are analysed qualitatively.

**Strengths:**

- To the best of my knowledge this method for combining RL with LLMs is novel. It also seems to be to be fairly general and I could imagine it being profitably applied to other domains.
- The method is well-described and sufficiently many algorithmic details are provided for it to be reproducible in future work.
- The method compares to strong, reasonable baselines and achieves state-of-the-art results in a well-designed round-robin tournament.
- The qualitative analysis is clear and interesting, providing insight into the capabilities of the agent.
- This paper (along with concurrent work by Xu et al.) establishes Werewolf as an interesting new evaluation domain that tests hitherto unexplored properties of LLM-based agents.

**Weaknesses:**

- Mischaracterisation of the Cicero algorithm. It is not the case that this algorithm defines arbitrary language actions and then chooses from these. Rather, the algorithm uses a large language model for open-ended policy-conditioned dialogue, and then uses an RL model to choose from the (game-predefined) actions. Therefore none of the authors' baselines are similar to Cicero. This weakness is mitigated by the fact that the authors' method is meaningfully different to Cicero in any case, but they should take care to characterise these differences more precisely.
- The human benchmarking results are rather weak, and some of the conclusions about robustness in this context feel like overclaiming to me. The real test of robustness would be to introduce one AI player in a game involving 6 other human players (as was done in the Cicero paper, for instance). Instead, the authors do the opposite, introducing one human into a game with 6 AIs. It it unclear whether this is a good test of robustness, because if the AIs play sufficiently out of distribution with respect to the human, it may be very hard for the human to have a sizeable degree of influence on the game. The authors should make it clearer earlier on in the paper that the robustness results are limited to the single-human setting, and discuss the limitations of this choice in the results section.
- There is no ethics statement accompanying this paper, yet the authors are developing agents which have the incentive to bluff humans and to collaborate against humans. While I strongly believe that this research should be conducted, and I believe it can be ethically justified from many angles, it is beholden on the authors to make these arguments. Please include an ethics statement in any future version of the paper.
- There are some missing literature citations that it would be good to include e.g. https://arxiv.org/abs/2305.19165, https://arxiv.org/abs/2303.00001.

**Questions:**

See "Weaknesses".

---

> ### Author Response · Authors · 2023-11-17
> **Rebuttal by Authors**
>
> Thank you for your appreciation of our work and thoughtful comments! We are heartened to see your recognition of our work’s novelty and the positive assessment of our experiment results. We hope our responses can address your concerns.
>
> **W1: Mischaracterisation of the Cicero algorithm.**
>
> Thank you for your explanation. However, we think we have the same understanding of Cicero as yours. In the second line on page 3 of our submission, we discussed the difference between Cicero and our method as follows.
>
> > The main difference between Cicero and our method is that they have a predefined atomic action set and select the action first to generate languages. By contrast, the actions in our methods are natural languages generated by LLMs during the game and our agent selects an action after these languages are produced.
>
> The "Actomic" baseline in Section 5.1 is similar to Cicero in that it also first uses an RL policy to choose from predefined actions and then uses the LLM to generate action-conditioned statements.
>
>
> **W2: Human benchmarking results and conclusions about robustness: The authors should make it clearer earlier on in the paper that the robustness results are limited to the single-human setting, and discuss the limitations of this choice in the results section.**
>
> We agree with your suggestions and have updated the Introduction and Experiment sections of our paper accordingly. In our case, since LLMs are pretrained on a large amount of human data, they can be regarded as human proxies and produces human-like behaviors [1]. We have provided example game logs of our agents in Appendix I to show that the evaluated agents’ behaviors are similar to humans’. Therefore, the single-human setting is a reasonable evaluation in our case. We agree with your comments on evaluation with multiple humans and have added a discussion on the limitation of single-human evaluation in Appendix H.4 of our revised paper.
>
>
> **W3: There is no ethics statement accompanying this paper.**
>
> We have added a section on the ethics and societal impact in our revised paper. Please see Appendix A for detailed discussions.
>
>
> **W4: There are some missing literature citations that it would be good to include.**
>
> Thank you for your suggestion. We have included these two citations in the Related Work section and discussed their relation to our method in our revised paper.
>
> ---
> We extend our sincere gratitude for your feedback and hope our answers have addressed your concerns. Thank you again for your support and appreciation of our work.

---

> ### Comment · Reviewer_Np2i · 2023-11-22
>
> I thank the authors for their response. A few points of clarification:
>
> *W1: mischaracterisation of the Cicero algorithm*.
>
> We do not believe that the authors have taken on board our advice. More specifically the following claim is false:
>
> "[Cicero selects] the action first to generate languages"
>
> It is not true that Cicero selects the action and only then generates language. In fact, Cicero interleaves action selection and language generation, and the action selection is conditioned on the previous conversation, as it must be for the conversation to be relevant to the strategy. The authors should update this statement in any camera-ready version.
>
> It is also not true that their "atomic" baseline is similar to Cicero. Their atomic baseline is a pure RL agent trained on atomic actions, whereas Cicero has both atomic actions and a conversational element. The authors should remove this statement in any camera-ready version.
>
> The difference between their agent and Cicero is therefore more subtle and should be clarified much more carefully. Specifically, in their agent, RL has as its action space the set of language utterances, whereas in Cicero, RL has as its action space the set of atomic actions in the game. In other words, they apply RL within the conversational component, which Cicero does not.
>
> *W2: Human benchmarking results and conclusions about robustness*.
>
> I disagree with the authors that their agents produce "human-like" behaviours. It is not sufficient to eyeball game transcripts as in Appendix I, since this is not a scientifically rigorous process. The justification that this evaluation is strong because the LLMs are "human-like" is simply not backed up with empirical evidence.
>
> It would be much better for the authors to point out that this is simply a starting point, and that it is important to evaluate in the multiple human case before drawing strong conclusions. In fact, pointing out a limitation in such a stark way is the hallmark of good science. As it is, the authors have weakened their paper by continuing to overclaim the significance of the single-human evaluation.
>
> Overall, I feel that the authors have failed to take on board my feedback, and these two concerns over their claims remain. If I could, I would lower my score by one point (two points seem too harsh). So please consider my new score to be a 7.

---

> > ### Author Response · Authors · 2023-11-22
> > **Author Response**
> >
> > Thank you for your clarification! We genuinely value your advice and apologize for not fully understanding your feedback in our previous reply. We have further revised our paper according to your advice and hope the following revisions can address your concerns.
> >
> >
> > **W1: mischaracterisation of the Cicero algorithm**
> >
> > We have revised our statement on Cicero in Section 2 (Related work) to make the difference between our agent and Cicero more clear and accurate. The revised statement is as follows.
> >
> > > The main difference between Cicero and our method is that Cicero uses the RL policy to choose from a predefined action set. By contrast, the actions in our methods are natural languages generated by LLMs during the game, and the RL policy is used to choose from these actions which are not known in advance.
> >
> > About the "atomic" baseline, we would like to clarify that it also takes the conversation history as input (using the embedding of information record and deduction result), and its action selection is conditioned on these previous conversations. However, we agree that it may be not appropriate to say the "atomic" agent is similar to Cicero and have revised our statement in Section 5.1 (Round-robin tournament) as follows.
> >
> > > The atomic agent predefines a set of high-level atomic actions and trains an RL policy with this fixed action space. The RL policy takes the embeddings of the information record and deduction result as input and selects the atomic action based on the game history. Then the natural language actions used in gameplay are generated by prompting the LLM to follow the selected atomic actions. In our case, the atomic action set consists of 13 actions including idle, target player_{0, ..., 6}, claim to be the {Werewolf, Seer, Doctor, Villager}, and do not reveal role.
> >
> >
> > **W2: Human benchmarking results and conclusions about robustness.**
> >
> > We have removed the overclaim on the significance of single-human evaluation and have rewritten the discussion on the limitation of single-human evaluation in Section 5.2 and Appendix H.4. The revised discussions are as follows.
> >
> > > (In Section 5.2 Human evaluation)
> > >
> > > The current single-human setting is a starting point to evaluate the robustness of our agent. A more comprehensive way is to play with multiple humans in one game and evaluate the performance of our agent. We discuss the limitations of single-human evaluation and future work on multi-human evaluation in Appendix H.4.
> >
> > > (In Appendix H.4 Limitations of single-human evaluation)
> > >
> > > Our human experiments in Section 5.2 serve as a starting point to evaluate the agents' robustness by letting one human play with six AI agents in one game. The limitation of this single-human evaluation is that if the policies of the AI agents are sufficiently out of the distribution of human policies, it may be very hard for the human players to have a reasonable influence on the game and could fail to be a good test of robustness. A more comprehensive way to evaluate robustness is to let one AI agent play with six human players in one game and compare the performance. This multi-human evaluation makes the game proceed in a manner more consistent with human behavior and is a better way to evaluate the robustness of the agents both as a teammate and as an opponent.
> >
> > ---
> > Please let us know if these revisions have effectively addressed your concerns. We are committed to making the necessary changes to clarify all aspects and are more than happy to take your further advice. Your support is invaluable to us and we would be very grateful if you could keep your score as an 8.

---

> > > ### Comment · Reviewer_Np2i · 2023-11-22
> > >
> > > Many thanks for your further reply.
> > >
> > > I am comfortable that you have now addressed my concerns, so I am happy to keep my score as an 8.

---

### Author Response · Authors · 2023-11-17
**We have updated our paper according to the reviews**

We would like to thank all reviewers for taking the time to review our submission and providing valuable feedback on our work. We have updated our paper and made the following changes according to your constructive comments. The changes are highlighted in blue in our revised paper.

1. **Introduction:** make it clear that the robustness results are in single-human settings.
2. **Related work:** discuss more relevant works including [1, 2, 3, 4].
3. **Experiment:**
    1. Round-robin tournament: train two more runs of RL policy and report both mean and std in Figure 3; add more explanation for Figure 3.
    2. Zero-shot transfer: report both mean and std in Table 1.
4. **Appendix A**: discussion on ethics and societal impact.
5. **Appendix B**: detailed rules of the game.
6. **Appendix D**: implementation details including policy architecture, reward design, and RL training hyperparameters.
7. **Appendix E**: additional experiments and ablation results.
8. **Appendix F**: quantitive results of emergent behaviors.
9. **Appendix G**: case study to show the benefit of using an RL policy.
10. **Appendix H**: discussion on application to other games, failure cases, and limitations.
11. **Appendix J**: example game log.

In response to the specific concerns and suggestions raised by each reviewer, we provide a detailed discussion for each of your reviews. We genuinely hope our response can provide a more comprehensive evaluation of our work and further reinforce the validity of our work.

Best regards,

Submission 7720 Authors

---
[1] Gandhi, Kanishk, Dorsa Sadigh, and Noah D. Goodman. "Strategic Reasoning with Language Models." arXiv preprint arXiv:2305.19165 (2023).

[2] Kwon, Minae, et al. "Reward design with language models." arXiv preprint arXiv:2303.00001 (2023).

[3] Wang, Shenzhi, et al. "Avalon's Game of Thoughts: Battle Against Deception through Recursive Contemplation." arXiv preprint arXiv:2310.01320 (2023).

[4] Guo, Jiaxian, et al. "Suspicion-Agent: Playing Imperfect Information Games with Theory of Mind Aware GPT-4." arXiv preprint arXiv:2309.17277 (2023).

---

### Author Response · Authors · 2023-11-20
**A Gentle Reminder**

Dear Reviewers,

Thank you once again for your valuable comments and suggestions. Your feedback has played a crucial role in enhancing both the quality and clarity of our paper.

While the discussion period is going to end in less than 3 days, we have not yet received the anticipated further responses. We remain eager to delve into the strengths and merits of our work with you. Your insights and suggestions are not only highly appreciated but also integral to our process, and we stand ready to make any necessary improvements to the paper.

We have also made further updates to our paper in Section 5.4 to intuitively show the benefit of using an RL policy. If you have any additional questions or require further clarification, please do not hesitate to reach out.

Best Regards,

Submission 7720 Authors

---

### Meta-Review · Area_Chair_kfUk · 2023-12-08

**Metareview:**

This paper proposes a novel LLM agent and the game Werewolf as a benchmark to evaluate LLM agents. The topic is very timely and led to active debate among the reviewers. Most remained in agreement that the proposed benchmark has significant potential for impact on the research community, but open questions remained about the proposed agent, its position in relation to other LLM agents and the current execution of the empirical evaluation. All of these could be addressed in future work to improve the paper's contribution and clarity.

**Justification For Why Not Higher Score:**

+ Open questions from all reviewers about if publication is sufficiently rigorous to merit publication at this time

**Justification For Why Not Lower Score:**

N/A

---

### Decision · Program_Chairs · 2024-01-16

Reject